# FIRST-ORDER OPTIMIZATION ALGORITHMS VIA DISCRETIZATION OF FINITE-TIME CONVERGENT FLOWS

## ABSTRACT

In this paper the performance of several discretization algorithms for two first-order finite-time optimization flows. These flows are, namely, the rescaled-gradient flow (RGF) and the signed-gradient flow (SGF), and consist of non-Lipscthiz or discontinuous dynamical systems that converge locally in finite time to the minima of gradient-dominated functions. We introduce three discretization methods for these first-order finite-time flows, and provide convergence guarantees. We then apply the proposed algorithms in training neural networks and empirically test their performances on three standard datasets, namely, CIFAR10, SVHN, and MNIST. Our results show that our schemes demonstrate faster convergences against standard optimization alternatives, while achieving equivalent or better accuracy.

## 1 INTRODUCTION

Consider the unconstrained minimization problem for a given cost function $f : \mathbb{R}^n \to \mathbb{R}$. When $f$ is sufficiently regular, the standard algorithm in continuous time (dynamical system) is given by

$$\dot{x} = F_{\mathrm{GF}}(x) \triangleq -\nabla f(x) \tag{1}$$

with $\dot{x} \triangleq \frac{\mathrm{d}}{\mathrm{d}t} x(t)$, known as the *gradient flow* (GF). Generalizing GF, the *q-rescaled* GF (q-RGF) Wibisono et al. (2016) given by

$$\dot{x} = -c \frac{\nabla f(x)}{\|\nabla f(x)\|_2^{\frac{q-2}{q-1}}} \tag{2}$$

with $c > 0$ and $q \in (1, \infty]$ has an asymptotic convergence rate $f(x(t)) - f(x^\star) = \mathcal{O}\left(\frac{1}{t^{q-1}}\right)$ under mild regularity, for $\|x(0) - x^\star\| > 0$ small enough, where $x^\star \in \mathbb{R}^n$ denotes a local minimizer of $f$. However, we recently proved Romero & Benosman (2020) that the q-RGF, as well as our proposed *q-signed* GF (q-SGF)

$$\dot{x} = -c \|\nabla f(x)\|_1^{\frac{1}{q-1}} \mathrm{sign}(\nabla f(x)), \tag{3}$$

where $\mathrm{sign}(\cdot)$ denotes the sign function, applied element-wise for (real-valued) vectors, are both finite-time convergent, provided that $f$ is gradient dominated of order $p \in (1, q)$. In particular, if $f$ is strongly convex, then q-RGF and q-SGF is finite-time convergent for any $q \in (2, \infty]$, since $f$ must be gradient dominated of order $p = 2$.

### CONTRIBUTION

In this paper, we explore three discretization schemes for the q-RGF (2) and q-SGF (3) and provide some convergence guarantees using results from hybrid dynamical control theory. In particular, we explore a forward-Euler discretization of RGF/SGF, followed by an explicit Runge-Kutta discretization, and finally a novel Nesterov-like discretization. We then test their performance on both synthetic and real-world data in the context of deep learning, namely, over the well-known datasets CIFAR10, SVHN, and MNIST.

### RELATED WORK

Propelled by the work of Wang & Elia (2011) and Su et al. (2014), there has been a recent and significant research effort dedicated to analyzing optimization algorithms from the perspective of

dynamical systems and control theory, especially in continuous time Wibisono et al. (2016); Wilson (2018); Lessard et al. (2016); Fazlyab et al. (2017b); Scieur et al. (2017); França et al. (2018); Fazlyab et al. (2018); Fazlyab et al. (2018); Taylor et al. (2018); França et al. (2019a); Orvieto & Lucchi (2019); Muehlebach & Jordan (2019). A major focus within this initiative is in *accceleration*, both in terms of trying to gain new insight into more traditional optimization algorithms from this perspective, or even to exploit the interplay between continuous-time systems and their potential discretizations for novel algorithm design Muehlebach & Jordan (2019); Fazlyab et al. (2017a); Shi et al. (2018); Zhang et al. (2018); França et al. (2019b); Wilson et al. (2019). Many of these papers also focus on explicit mappings and matchings of convergence rates from the continuous-time domain into discrete time.

For older work connecting ordinary differential equations (ODEs) and their numerical analysis, with optimization algorithms, see Botsaris (1978a;b); Zghier (1981); Snyman (1982; 1983); Brockett (1988); Brown (1989). In Helmke & Moore (1994), the authors studied relationships between linear programming, ODEs, and general matrix theory. Further, Schropp (1995) and Schropp & Singer (2000) explored several aspects linking nonlinear dynamical systems to gradient-based optimization, including nonlinear constraints.

Tools from Lyapunov stability theory are often employed for this purpose, mainly because there already exists a rich body of work within the nonlinear systems and control theory community for this purpose. In particular, typically in previous works, one seeks asymptotically Lyapunov stable gradient-based systems with an equilibrium (stationary point) at an isolated extremum of the given cost function, thus certifying local convergence. Naturally, global asymptotic stability leads to global convergence, though such an analysis will typically require the cost function to be strongly convex everywhere.

For physical systems, a Lyapunov function can often be constructed from first principles via some physically meaningful measure of energy (*e.g.*, total energy = potential energy + kinetic energy). In optimization, the situation is somewhat similar in the sense that a suitable Lyapunov function may often be constructed by taking simple surrogates of the objective function as candidates. For instance, $V(x) \triangleq f(x) - f(x^\star)$ can be a good initial candidate. Further, if $f$ is continuously differentiable and $x^\star$ is an isolated stationary point, then another alternative is $V(x) \triangleq \|\nabla f(x)\|^2$. However, most fundamental and applied research conducted in systems and control regarding Lyapunov stability theory deals exclusively with *continuous-time* systems. Unfortunately, (dynamical) stability properties are generally not preserved for simple forward-Euler and sample-and-hold discretizations and control laws Stuart & Humphries (1998). Furthermore, practical implementations of optimization algorithms in modern digital computers demand discrete-time. Nonetheless, it has been extensively noted that a vast amount of general Lyapunov-based results appear to have a discrete-time equivalent.

In that sense, we aim here to start from the $q$-RGF, and $q$-SGF continuous flows, characterized by their Lyapunov-based finite-time convergence, and seek discretization schemes, which allow us to 'shadow' the solutions of these flows in discrete time, hoping to achieve an acceleration of the discrete methods inspired from the finite-time convergence characteristics of the continuous flows.

## 2 Optimization Algorithms as Discrete-Time Systems

Generalizing (1), (2), and (3), consider a continuous-time algorithm (dynamical system) modeled via an ordinary differential equation (ODE)

$$\dot{x} = F(x) \tag{4}$$

for $t \geq 0$, or, more generally, a differential inclusion

$$\dot{x}(t) \in \mathcal{F}(x(t)) \tag{5}$$

a.e. $t \geq 0$ (*e.g.* for the $q = \infty$ case), such that $x(t) \to x^\star$ as $t \to t^\star$. In the case of the $q$-RGF (2) and $q$-SGF (3) for $f$ gradient dominated of order $p \in (1, q)$, we have finite-time convergence, and thus $t^\star = t^\star(x(0)) < \infty$.

Most of the popular numerical optimization schemes can be written in a state-space form (*i.e.*, recursively), as

$$X_{k+1} = F_{\mathrm{d}}(k, X_k) \tag{6a}$$

$$x_k = G(X_k) \tag{6b}$$

for $k \in \mathbb{Z}_+ \triangleq \{0, 1, 2, \ldots\}$ and a given $X_0 \in \mathbb{R}^m$ (typically $m \geq n$), where $F_{\mathrm{d}} : \mathbb{Z}_+ \times \mathbb{R}^m \to \mathbb{R}^n$ and $G : \mathbb{R}^m \to \mathbb{R}^n$.

Naturally, (6) can be seen as a discrete-time dynamical system constructed by discretizing (4) in time. In particular, we have $x_k \approx x(t_k)$, where $\{0 = t_0 < t_1 < t_2 < \ldots\}$ denotes a time partition and $x(\cdot)$ a solution to (4) or (5) as appropriate. Therefore, we call $X_k$ and $x_k$, respectively, the *state* and *output* at time step $k$. Whenever $F_{\mathrm{d}}(k, X)$ does not depend on $k$, we will drop $k$ and thus write $F_{\mathrm{d}}(X)$ instead. Whenever $m = n$, we will denote take $G(X) \triangleq X$ and replace $X$ and $X_k$ by $x$ and $x_k$, respectively.

**Example 1.** The standard gradient descent (GD) algorithm

$$x_{k+1} = x_k - \eta \nabla f(x_k) \tag{7}$$

with step size (learning rate) $\eta > 0$ can be readily written in the form (6) by taking $m = n$, $F_{\mathrm{d}}(x) \triangleq x - \eta \nabla f(x)$, and $G(x) \triangleq x$.

- If the step sizes are adaptive, *i.e.* if we replace $\eta$ by a sequence $\{\eta_k\}$ with $\eta_k > 0$, then we only need to replace $F_{\mathrm{d}}(k, x) \triangleq x - \eta_k \nabla f(x)$, provided that $\{\eta_k\}$ is not computed using feedback from $\{x_k\}$ (*e.g.* through a line search method).

- If we do wish to use *feedback*[1] (and no memory past the most recent output and step size), then we can set $m = n + 1$, $G([x; \eta]) \triangleq x$, and $F_{\mathrm{d}}([x; \eta]) \triangleq [F_{\mathrm{d}}^{(1)}([x; \eta]); F_{\mathrm{d}}^{(2)}([x; \eta])]$, where $F_{\mathrm{d}}^{(1)}([x; \eta]) \triangleq x - \eta \nabla f(x)$, and $F_{\mathrm{d}}^{(2)}$ is a user-defined function that dictates the updates in the step size. In particular, an open-loop (no feedback) adaptive step size $\{\eta_k\}$ may also be achieved under this scenario, provided that it is possible to write $\eta_{k+1} = F_{\mathrm{d}}^{(2)}(\eta_k)$. If this is not possible (and still open-loop step size), then we may take $F_{\mathrm{d}}^{(2)}(k, X) \triangleq \eta_{k+1}$, and of course add a $k$-argument in $F_{\mathrm{d}}$.

- If we wish to use individual step sizes for each the $n$ components of $\{x_k\}$, then it suffices to take $\eta_k$ as an $n$-dimensional vector (thus $m = 2n$), and make appropriate changes in $F_{\mathrm{d}}$ and $G$.

In each of these cases, GD can be seen as a forward-Euler discretization of the GF (1), *i.e.*,

$$x_{k+1} = x_k + \Delta t_k F_{\mathrm{GF}}(x_k) \tag{8}$$

with $F_{\mathrm{GF}} = -\nabla f$ and adaptive time step $\Delta t_k \triangleq t_{k+1} - t_k$ chosen as $\Delta t_k = \eta_k$.

**Example 2.** The proximal point algorithm (PPA)

$$x_{k+1} = \arg\min_{x \in \mathbb{R}^n} \left\{ f(x) + \frac{1}{2\eta_k} \|x - x_k\|^2 \right\} \tag{9}$$

with step size $\eta_k > 0$ (open loop, for simplicity) can also be written in the form (6), by taking $m = n$, $F_{\mathrm{d}}(k, x) \triangleq \arg\min_{x' \in \mathbb{R}^n} \{ f(x') + \frac{1}{2\eta_k} \|x' - x\|^2 \}$, and $G(x) \triangleq x$. Naturally, we need to assume sufficient regularity for $F_{\mathrm{d}}(k, x)$ to exist and we must design a consistent way to choose $F_{\mathrm{d}}(k, x)$ when multiple minimizers exist in the definition of $F_{\mathrm{d}}(k, x)$. Alternatively, these conditions must be satisfied, at the very least, at every $(k, x) \in \{(0, x_0), (1, x_1), (2, x_2), \ldots\}$ for a particular chosen initial $x_0 \in \mathbb{R}^n$.

Assuming sufficient regularity, we have $\nabla_x \{ f(x) + \frac{1}{2\eta_k} \|x - x_k\|^2 \}|_{x=x_{k+1}} = 0$, and thus

$$\nabla f(x_{k+1}) + \frac{1}{\eta_k}(x_{k+1} - x_k) = 0 \iff x_{k+1} = x_k + \Delta t_k F_{\mathrm{GF}}(x_{k+1}) \tag{10}$$

with $\Delta t_k = \eta_k$, which is precisely the backward-Euler discretization of the GF (1).

---

[1] Also known as *closed-loop* design in control-theoretic and reinforcement learning terminology, meaning that $\eta_k = \varphi(k, x_k)$ for some $\varphi : \mathbb{Z}_+ \times \mathbb{R}^n \to \mathbb{R}_+$ that does not depend on $\{X_0, X_1, X_2, \ldots\}$. On the other hand, *open-loop* design can be seen as closed loop with $\varphi(k, \cdot)$ constant for each $k \in \mathbb{Z}_+$.

**Example 3.** The Nesterov accelerated gradient descent (N-AGD)

$$y_k = x_k + \beta_k(x_k - x_{k-1}) \tag{11a}$$

$$x_{k+1} = y_k - \eta_k \nabla f(y_k) \tag{11b}$$

with step size $\eta_k > 0$ and momentum coefficient $\beta_k > 0$ (both open loop, for simplicity), can be written in the form (6) by taking $m = 2n$,

$$F_{\mathrm{d}}(k, [y; x]) \triangleq \begin{bmatrix} (1 + \beta_{k+1})(y - \eta_k \nabla f(y)) - \beta_{k+1} x \\ y - \eta_k \nabla f(y) \end{bmatrix} \tag{12}$$

and $G([y; x]) \triangleq x$ for $y, x \in \mathbb{R}^n$. In other words, $X_k = [y_k; x_k]$. Traditionally, $\beta_k = \frac{k-1}{k-2}$, but clearly, if we set $\eta_k = \eta > 0$ and $\beta_k = \beta \in (0, 1)$ (in practice, $\eta \approx 0$ and $\beta \approx 1$), then we can drop $k$ from $F_{\mathrm{d}}(k, [y; x])$.

There exist a few approaches in the literature on the interpretation of N-AGD (11b) as the discretization of a second-order continuous-time dynamical system, namely via a vanishing step size argument Su et al. (2014), or via symplectic Euler schemes of crafted Hamiltonian systems Muehlebach & Jordan (2019); França et al. (2019b).

## 3 Proposed Algorithms via Discretization

In this section, we propose three classes of optimization algorithms via discretization of the $q$-RGF (2) and $q$-SGF (3). But first, we review the necessary conditions to ensure finite-time convergence of these flows.

Given $q \in (1, \infty]$, let $F_{q-\mathrm{RGF}}(x)$ and $F_{q-\mathrm{SGF}}(x)$ be defined, respectively, by the RHS of (2) and (3). The hyperparameter $c > 0$ is not explicitly denoted in $F_{q-\mathrm{RGF}}, F_{q-\mathrm{SGF}}$. Next, borrowing terminology from Wilson et al. (2019), we say that $f$ (assumed continuously differentiable) is *$\mu$-gradient dominated of order $p \in (1, \infty]$* (with $\mu > 0$) near the local minimizer $x^\star$ if

$$\frac{p-1}{p} \|\nabla f(x)\|^{\frac{p}{p-1}} \geq \mu^{\frac{1}{p-1}} (f(x) - f^\star) \tag{13}$$

for every $x \in \mathbb{R}^n$ near $x = x^\star$, where $f^\star = f(x^\star)$. When $\mu > 0$ is unknown or unimportant, but known to exist, we will omit it in the previous definition. It can be proved that continuously differentiable strongly convex functions are gradient dominated of order $p = 2$. Furthermore, if $f$ is gradient dominated (of any order) w.r.t. $x^\star$, then $x^\star$ is an isolated stationary point of $f$.

**Remark 1.** For strongly convex functions, gradient dominance of order $p = 2$ can be established. In fact, gradient dominance is usually defined exclusively for order $p = 2$, often referred to as the Polyak-Łojasiewicz (PL) inequality, which was introduced by Polyak (1963) to relax the (strong) convexity assumption commonly used to show convergence of the GD algorithm (7). The PL inequality can also be used to relax convexity assumptions of similar gradient and proximal-gradient methods Karimi et al. (2016); Attouch & Bolte (2009). Our adopted generalized notion of gradient dominance is strongly tied to the Łojasiewicz gradient inequality from real analytic geometry, established by Łojasiewicz (1963; 1965)[2] independently and simultaneously from Polyak (1963), and generalizing the PL inequality. More precisely, this inequality is typically written as

$$\|\nabla f(x)\| \geq C \cdot |f(x) - f^\star|^\theta \tag{14}$$

for every $x \in \mathbb{R}^n$ in a small enough open neighborhood of the stationary point $x = x^\star$, for some $C > 0$ and $\theta \in \left(\frac{1}{2}, 1\right]$. This inequality is guaranteed for analytic functions Łojasiewicz (1965). More precisely, when $x^\star$ is a local minimizer of $f$, the aforementioned relationship is explicitly given by

$$C = \left(\frac{p}{p-1}\right)^{\frac{p-1}{p}} \mu^{\frac{1}{p}}, \qquad \theta = \frac{p-1}{p}. \tag{15}$$

Therefore, analytic functions are always gradient dominated. However, while analytic functions are always smooth, smoothness is not required to attain gradient dominance.

---

[2]For more modern treatments in English, see Łojasiewicz & Zurro (1999); Bolte et al. (2007)

We are now ready to state the finite-time convergence of the $q$-RGF (2) and $q$-SGF (3).

**Theorem 1.** *Romero & Benosman (2020) Suppose that $f : \mathbb{R}^n \to \mathbb{R}$ is continuously differentiable and $\mu$-gradient dominated of order $p \in (1, \infty)$ near a strict local minimizer $x^\star \in \mathbb{R}^n$. Let $c > 0$ and $q \in (p, \infty]$. Then, any maximal solution $x(\cdot)$, in the sense of Filippov, to the $q$-RGF (2) or $q$-SGF (3) will converge in finite time to $x^\star$, provided that $\|x(0) - x^\star\| > 0$ is sufficiently small. More precisely, $\lim_{t \to t^\star} x(t) = x^\star$, where the convergence time $t^\star < \infty$ may depend on which flow is used, but in both cases is upper bounded by*

$$t^\star \leq \frac{\|\nabla f(x_0)\|^{\frac{1}{\theta} - \frac{1}{\theta'}}}{c C^{\frac{1}{\theta}} \left(1 - \frac{\theta}{\theta'}\right)}, \tag{16}$$

*where $x_0 = x(0)$, $C = \left(\frac{p}{p-1}\right)^{\frac{p-1}{p}} \mu^{\frac{1}{p}}$, $\theta = \frac{p-1}{p}$, and $\theta' = \frac{q-1}{q}$. In particular, given any compact and positively invariant subset $S \subset \mathcal{D}$, both flows converge in finite time with the aforementioned convergence time upper bound (which can be tightened by replacing $\overline{\mathcal{D}}$ with $S$) for any $x_0 \in S$. Furthermore, if $\mathcal{D} = \mathbb{R}^n$, then we have global finite-time convergence, i.e. finite-time convergence to any maximal solution (in the sense of Filippov) $x(\cdot)$ with arbitrary $x_0 \in \mathbb{R}^n$.*

In essence, the analysis (introduced in Romero & Benosman (2020)) consists of leveraging the gradient dominance to show that the energy function $\mathcal{E}(t) \triangleq f(x(t)) - f^\star$ satisfies the Lyapunov-like differential inequality $\dot{\mathcal{E}}(t) = \mathcal{O}(\mathcal{E}(t)^\alpha)$ for some $\alpha < 1$. The detailed proof is recalled in Appendix C for completeness.

### 3.1 MAIN RESULT 1: DISCRETIZATION ALGORITHMS AND THEIR CONVERGENCE ANALYSIS

#### 3.1.1 FORWARD-EULER DISCRETIZATION

First, we propose a simple forward Euler discretization of the finite-time convergent flows

$$x_{k+1} = x_k + \eta F(x_k), \; \eta > 0 \tag{17}$$

where $F \in \{F_{q-\mathrm{RGF}}, F_{q-\mathrm{SGF}}\}$. We show later, in Theorem 2, that this simple method leads, for small enough $\eta > 0$, to solutions that are $\epsilon$-close to the finite-time flows.

#### 3.1.2 EXPLICIT RUNGE-KUTTA DISCRETIZATION

We propose to use the following discretization

$$x_{k+1} = x_k + \eta \sum_{i=1}^{K} \alpha_i F(y_k^i), \; y_k^1 = x_k, \; \sum_{i=1}^{K} \alpha_i = 1, \tag{18a}$$

$$y_k^i = x_k + \eta \sum_{j=1}^{i-1} \beta_j F(y^j), i > 1, \tag{18b}$$

for $\eta > 0$, $K \in \{1, 2, 3, \ldots\}$, and $F \in \{F_{q-\mathrm{RGF}}, F_{q-\mathrm{SGF}}\}$. This method is well-known to be numerically stable under the consistency condition $\sum_{i=1}^{i=K} \alpha_i = 1$, Stuart & Humphries (1996). However, in our optimization framework, we want to be able to guarantee that the stable numerical solution of (18) remains close to the solution of the continuous flows. In other words, we seek arbitrarily small global error, also known as *shadowing*. This will be discussed in Theorem 2.

#### 3.1.3 NESTEROV-LIKE DISCRETIZATION

First, we rewrite Nesterov's accelerated GD as

$$x_{k+1} = x_k + \beta y_k - \eta \nabla f(x_k + \beta y_k) \tag{19a}$$
$$y_{k+1} = x_{k+1} - x_k, \tag{19b}$$

where $y_k$ now serves as a momentum term. We argue that Nesterov's acceleration can be interpreted as actually applying the discretization given by (19) to the GF (1), i.e., by seeing the term $-\eta \nabla f(x_k + \beta y_k)$ as a mapping applied to the GF flow (1) at $x_k + \beta y_k$, as $-\eta F_{GF}(x_k + \beta y_k)$.

Therefore, given any optimization flow represented by the continuous-time system $\dot{x} = F(x)$, locally convergent to a local minimizer $x^\star \in \mathbb{R}^n$ of a cost function $f : \mathbb{R}^n \to \mathbb{R}$, we can replicate Nesterov's acceleration of (1). More precisely, we obtain the algorithm

$$x_{k+1} = x_k + \eta F(x_k + \beta y_k) + \beta y_k \tag{20a}$$

$$y_{k+1} = x_{k+1} - x_k. \tag{20b}$$

Based on this idea, we propose two 'Nesterov-like' discrete optimization algorithms. The first one based on the $q$-RGF continuous flow, is defined as:

$$x_{k+1} = x_k + \eta\Big( - c\frac{\nabla f(x_k + \beta y_k)}{\|\nabla f(x_k + \beta y_k)\|^{\frac{q-2}{q-1}}}\Big) + \beta y_k \tag{21a}$$

$$y_{k+1} = x_{k+1} - x_k. \tag{21b}$$

The second algorithm is based on the $q$-SGF continuous flow, and is given by:

$$x_{k+1} = x_k + \eta\Big( - c\|\nabla f(x_k + \beta y_k)\|_1^{\frac{1}{q-1}} \operatorname{sign}(\nabla f(x_k + \beta y_k)))\Big) + \beta y_k \tag{22a}$$

$$y_{k+1} = x_{k+1} - x_k. \tag{22b}$$

### 3.1.4 CONVERGENCE ANALYSIS

We present here some convergence results of the three proposed discretizations. The analysis summarized in Theorem 2 is based on tools from hybrid control theory, and is detailed in Appendix D[3].

**Theorem 2.** *Suppose that $f : \mathbb{R}^n \to \mathbb{R}$ is continuously differentiable, locally $L_f$-Lipschitz, and $\mu$-gradient dominated of order $p \in (2, \infty)$ in a compact neighborhood $S$ of a strict local minimizer $x^\star \in \mathbb{R}^n$. Let $c > 0$ and $q \in (p, \infty]$. Then, for a given initial condition $x_0 \in S$ any maximal solution $x(t)$, $x(0) = x_0$, (in the sense of Filippov) to the q-RGF given by (2) or the q-SGF flow (3), there exists an arbitrarily small $\epsilon > 0$ such that the solution $x_k$ of any of the discrete algorithms (17), (18), (21), or (22), with sufficiently small $\eta > 0$, are $\epsilon$-close to $x(t)$, i.e., $\|x_k - x\| \le \epsilon$, and s.t. the following convergence bound holds*

$$\|f(x_k) - f(x^\star)\| \le L_f\epsilon + [(f(x_0) - f(x^\star))^{(1-\alpha)} - \tilde{c}(1-\alpha)\eta k]^{1/(1-\alpha)}, \ L_f > 0, \ k \le k^\star, \tag{23}$$

*where $\alpha = \frac{\theta}{\theta'}$, $\theta = \frac{p-1}{p}$, $\theta' = \frac{q-1}{q}$, $\tilde{c} = c\left(\left(\frac{p}{p-1}\right)^{\frac{p-1}{p}} \mu^{\frac{1}{p}}\right)^{\frac{1}{\theta'}}$, and $k^\star = \frac{(f(x_0)-f(x^\star))^{(1-\alpha)}}{\tilde{c}(1-\alpha)\eta}$.*

Theorem 2 thus shows that that $\epsilon$-convergence of $x_k \to x^\star$ can be achieved in a finite number of steps upper bounded by $k^\star = \frac{(f(x_0)-f(x^\star))^{(1-\alpha)}}{\tilde{c}(1-\alpha)\eta}$. This is a preliminary convergence result, which is meant to show the existence of discrete solutions obtained via the proposed discretization algorithms, which are $\epsilon$-close to the continuous solutions of the finite-time flows. We also underline here that after $x_k$ reaches an $\epsilon$-neighborhood of $x^\star$, then $x_{k+1} \approx x_k$, $\forall k > k^\star$, since $x^\star$ is an equilibrium point of the continuous flows; see Definition 2 in Appendix B.

## 4 MAIN RESULT 2: NUMERICAL EXPERIMENTS

### 4.1 NUMERICAL TESTS ON AN ACADEMIC EXAMPLE

Let us show first on a simple numerical example that the acceleration in convergence, proven in continuous time for certain range of the hyperparmeters, can translate to some convergence acceleration in discrete time, as shown in Theorem 2. We consider the Rosenbrock function $f : \mathbb{R}^2 \to \mathbb{R}$, given by $f(x_1, x_2) = (a - x_1)^2 + b(x_2 - x_1^2)^2$, with parameters $a, b \in \mathbb{R}$. This function admits exactly one stationary point $(x_1^\star, x_2^\star) = (a, a^2)$ for $b \ge 0$, and is locally strongly convex, hence locally satisfies gradient dominance of order $p = 2$, which allows us to select $q > 2$ in $q$-RGF

---

[3]Note that there might be several ways of approaching this proof. For instance, one could follow the general results on stochastic approximation of set-valued dynamical systems, using the notion of perturbed solutions to differential inclusions presented in Benaïm et al. (2005).

and $q$-SGF to achieve finite-time convergence in continuous-time. We report in Figure 1 the mean performance of all three discretizations for $q$-RGF and $q$-SGF[4] with fixed step size[5], for several values of $q$, for 10 random initial conditions in $[0, 2]$. We observe for all three discretizations that, as expected from the continuous flow analysis, for $q$ close to 2, $q$-RGF behaves similar to GD in terms of convergence rate, whereas for $q > 2$ the finite-time convergence in continuous time seems to translate to some acceleration in this simple discretization method. Similarly for $q$-SGF, $q$ closer to 2 translates to less accelerated algorithms, with a behavior similar to GD, whereas larger $q$ values lead to accelerated convergence.

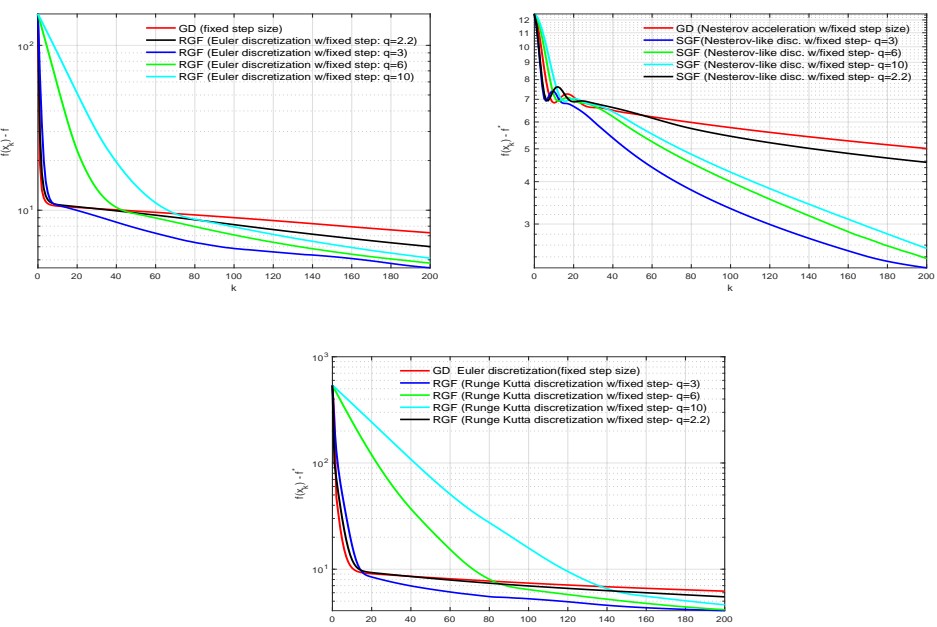

Figure 1: Example of the proposed discretization algorithms of finite-time $q$-RGF and $q$-SGF

## 4.2 NUMERICAL EXPERIMENTS ON REAL-WORLD DATA

We report here the results of our experiments using deep neural network (DNN) training on three well-known datasets, namely, CIFAR10, MNIST, and SVHN. We report results on CIFAR10, and SVHN in the sequel, while results on MNIST can be found in Appendix E. Note that, we use Pytorch platform to conduct all the tests reported in this paper, and do not use GPUs. We underline here that the DNNs are non-convex globally, however, one could assume at least local convexity, hence local gradient dominance of order $p = 2$, thus, we will select $q > 2$ in our experiments (see (Remark 2, Appendix E) for more explanations on the choice of $q$).

### 4.2.1 EXPERIMENT ON CIFAR10

In this experiment, we use the proposed algorithms to train a VGG16 CNN model with cross entropy loss, e.g., Simonyan & Zisserman (2015) on the CIFAR10 dataset. We divided the dataset into a training set spread in 50 batches of 1000 images each, and a test set of 10 batches with 1000 images each. We ran 20 epochs of training over all the training batches. Since Nesterov accelerated GD is one of the most efficient methods is DNN applications, to conduct fair comparisons we implemented our Nesterov-like discretization of $q$-RGF ($c = 1$, $q = 3$, $\eta = 0.04$, $\mu = 0.9$), and the Nesterov-like discretization of $q$-SGF ($c = 10^{-3}$, $q = 3$, $\eta = 0.04$, $\mu = 0.9$). We compare against the

---

[4]To avoid overloading the figures we had to choose one flow at a time, either $q$-RGF or $q$-SGF. More results can be found in Appendix E

[5]We did multiple iterations to find the best step size for each algorithm (best values where between $10^{-4}$ and $10^{-2}$ depending on the algorithm). Details of the step size for each test are given in Appendix E.

mainstream algorithms[6], such as, Nesterov's accelerated gradient descent (GD), Adam, Adaptive gradient (AdaGrad), per-dimension learning rate method for gradient descent (AdaDelta), and Root Mean Square Propagation (RMSprop)[7]. Note that, all algorithms have been implemented in their stochastic version[8], i.e., using mini-batches implementation, with constant step size. For instance, in Figures 2, 3[9], we see the training loss for the different optimization algorithms. We notice that the proposed Algorithms, RGF, and SGF, quickly separate from the GD, and RMS algorithms in terms of convergence speed, but also ends up with an overall better performance on the test set $84\%$ vs. $83\%$ for GD, and RMS. We also note that other algorithms, such as, AdaGrad and AdaDelta behave similarly to RGF in terms of convergence speed, but lack behind in terms of final performance $75\%$ and $68\%$, respectively. Finally, in Figure 3, we notice that Adam is slower in terms of computation time w.r.t SGF, and RGF, with an average lag of $8\,\text{min}$, and $80\,min$, respectively. However, to be fair one, has to underline that Adam is an adaptive method based on the extra computations and memory overhead of lower order moments estimate and bias correction, Kingma & Ba (2015). Furthermore, to better compare the performance of these algorithms, we report in Figure 2 the loss on the test dataset over the learning iterations. We confirm that RGF and SGF performs better than GD, RMSprop, and Adam, while avoiding the overfitting observed with AdaGrad and AdaDelta.

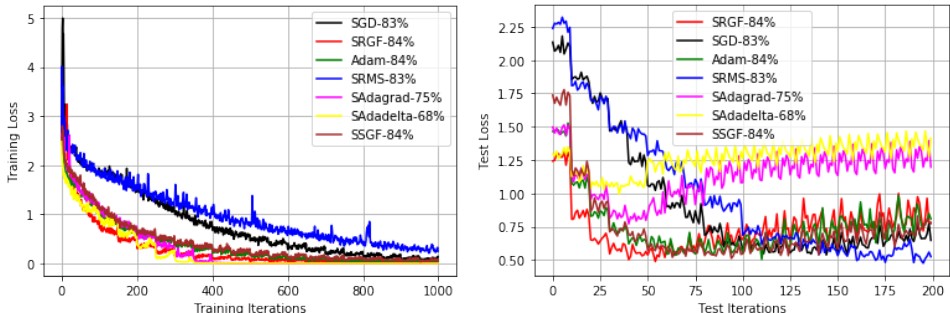

Figure 2: Losses for several optimization algorithms- VGG-16- CIFAR10: Train loss (left), test loss (right)- We add an S before the name of an algorithm to denote its stochastic implementation.

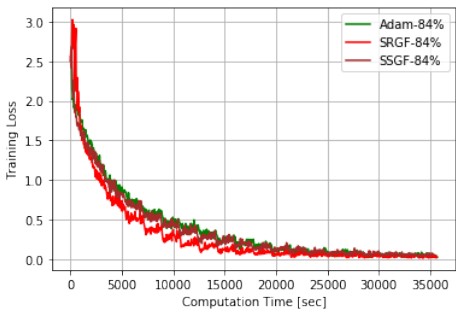

Figure 3: Training loss vs. computation time for several optimization algorithms- VGG-16- CIFAR10

---

[6]We run several tests by trying to optimally tune the parameters of each algorithms on a validation set (tenth of training set), and we are reporting the best final accuracy performance we could obtain for each one. We have implemented all algorithms with the same total number of iterations, so that we can compare the convergence speed of all algorithms against a common iteration range. Details of the hyper-parameter values are given in Appendix.

[7]Original reference for each method can be found in: https://pytorch.org/docs/stable/optim.html

[8]We want to underline here that our first tests were done in the deterministic setting, however, to compare the propsed optimization methods against the best optimization algorithms available for DNNs training, we decided to also conduct comparison tests in the stochastic setting. Since the results remain qualitatively unchanged, we only report here the results due to the stochastic setting.

[9]To avoid overloading the figures we reported only the computation time plots of the three most competitive methods: RGF, SGF and Adam.

### 4.2.2 EXPERIMENTS ON SVHN DATASET

We test the proposed algorithms to train the same VGG16 CNN model with cross entropy loss on the SVHN dataset. We divided the dataset into a training set of 70 batches with 1000 images each, and a test set of 10 of 1000 images each, and ran 20 epochs of training over all the training batches. We tested the Nesterov-like discretization of $q$-RGF ($c = 1$, $q = 3$, $\eta = 0.04$, $\mu = 0.09$), and the Nesterov-like discretization of $q$-SGF ($c = 10^{-3}$, $q = 11$, $\eta = 0.04$, $\mu = 0.09$) against Nesterov's accelerated gradient descent (GD), and Adam[10]. Note from Figures 4, 5 it is clear that RGF and SGF give a good performance in terms of convergence speed, and final test performance $93\%$. We can also observe in Figure 5 that SGF, RGF are faster than GD, and all three methods are faster (in average $41\ min$ for GD, $58\ min$ for SGF, $75\ min$ for RGF) than Adam as expected since it is an adaptive scheme with more computation steps (see our discussion of Adam in Section 4.2.1). More

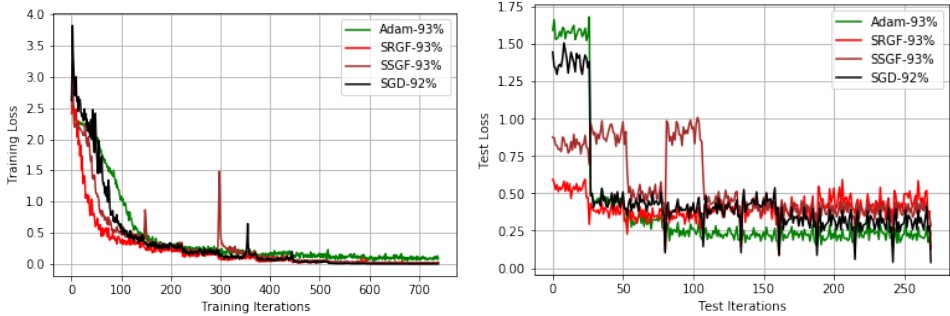

Figure 4: Losses for several optimization algorithms - CNN- SVHN: Train loss (left), test loss (right)

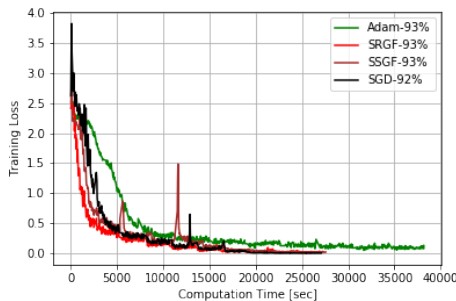

Figure 5: Training loss vs. computation time for several optimization algorithms- VGG-16- SVHN

numerical results on MNIST, and on SVHN using Euler and Runge-Kutta descretization, showing similar qualitative results, can be found in Appendix E.

## 5 CONCLUSION

We studied connections between optimization algorithms and continuous-time representations (dynamical systems) via discretization. We then reviewed two families of non-Lipschitz or discontinuous first-order optimization flows for continuous-time optimization, namely the $q$-RGF and $q$-SGF, whose distinguishing characteristic is their finite-time convergence. We then proposed three discretization methods for these flows, namely a forward-Euler discretization, followed by an explicit Runge-Kutta discretization, and finally a novel Nesterov-like discretization. Based on tools from hybrid systems control theory, we proved a convergence bound for these algorithms. Finally, we conducted numerical experiments on known deep neural net benchmarks, which showed that the proposed discrete algorithms can outperform some state of the art algorithms, when tested on large DNN models.

---

[10]We also tested Adaptive gradient (AdaGrad), per-dimension learning rate method for gradient descent (AdaDelta), and Root Mean Square Propagation (RMSprop). However, since their performance was not competitive we decided not to report the plots to avoid overloading the figures.

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

## A   DISCONTINUOUS SYSTEMS AND DIFFERENTIAL INCLUSIONS

[11] Recall that for an initial value problem (IVP)

$$\dot{x}(t) = F(x(t)) \tag{24a}$$
$$x(0) = x_0 \tag{24b}$$

with $F : \mathbb{R}^n \to \mathbb{R}^n$, the typical way to check for existence of solutions is by establishing continuity of $F$. Likewise, to establish uniqueness of the solution, we typically seek Lipschitz continuity. When $F$ is discontinuous, we may understand (24a) as the Filippov differential inclusion

$$\dot{x}(t) \in \mathcal{K}[F](x(t)), \tag{25}$$

where $\mathcal{K}[F] : \mathbb{R}^n \rightrightarrows \mathbb{R}^n$ denotes the Filippov set-valued map given by

$$\mathcal{K}[F](x) \triangleq \bigcap_{\delta > 0} \bigcap_{\mu(S) = 0} \overline{\text{co}}\, F(B_\delta(x) \setminus S), \tag{26}$$

where $\mu$ denotes the usual Lebesgue measure and $\overline{\text{co}}$ the convex closure, i.e. closure of the convex hull $\overline{\text{co}}$. For more details, see Paden & Sastry (1987). We can generalize (25) to the differential inclusion Bacciotti & Ceragioli (1999)

$$\dot{x}(t) \in \mathcal{F}(x(t)), \tag{27}$$

where $\mathcal{F} : \mathbb{R}^n \rightrightarrows \mathbb{R}^n$ is some set-valued map.

**Definition 1** (Carathéodory/Filippov solutions). We say that $x : [0, \tau) \to \mathbb{R}^n$ with $0 < \tau \leq \infty$ is a *Carathéodory solution* to (27) if $x(\cdot)$ is absolutely continuous and (27) is satisfied a.e. in every compact subset of $[0, \tau)$. Furthermore, we say that $x(\cdot)$ is a *maximal* Carathéodory solution if no other Carathéodory solution $x'(\cdot)$ exists with $x = x'|_{[0,\tau)}$. If $\mathcal{F} = \mathcal{K}[F]$, then Carathéodory solutions are referred to as *Filippov solutions*.

For a comprehensive overview of discontinuous systems, including sufficient conditions for existence (Proposition 3) and uniqueness (Propositions 4 and 5) of Filippov solutions, see the work of Cortés (2008). In particular, it can be established that Filippov solutions to (24) exist, provided that the following assumption (Assumption 1) holds.

**Assumption 1** (Existence of Filippov solutions). $F : \mathbb{R}^n \to \mathbb{R}^n$ is defined almost everywhere (a.e.) and is Lebesgue-measurable in a non-empty open neighborhood $U \subset \mathbb{R}^n$ of $x_0 \in \mathbb{R}^n$. Further, $F$ is locally essentially bounded in $U$, *i.e.*, for every point $x \in U$, $F$ is bounded a.e. in some bounded neighborhood of $x$.

More generally, Carathéodory solutions to (27) exist (now with arbitrary $x_0 \in \mathbb{R}^n$), provided that the following assumption (Assumption 2) holds.

**Assumption 2** (Existence of Carathéodory solutions). $\mathcal{F} : \mathbb{R}^n \rightrightarrows \mathbb{R}^n$ has nonempty, compact, and convex values, and is *upper semi-continuous*.

Filippov & Arscott (1988) proved that, for the Filippov set-valued map $\mathcal{F} = \mathcal{K}[F]$, Assumptions 1 and 2 are equivalent (with arbitrary $x_0 \in \mathbb{R}^n$ in Assumption 1).

Uniqueness of the solution requires further assumptions. Nevertheless, we can characterize the Filippov set-valued map in a similar manner to Clarke's generalized gradient, as seen in the following proposition.

**Proposition 1** (Theorem 1 of Paden & Sastry (1987)). *Under Assumption 1, we have*

$$\mathcal{K}[F](x) = \left\{ \lim_{k \to \infty} F(x_k) : x_k \in \mathbb{R}^n \setminus (\mathcal{N}_F \cup S) \text{ s.t. } x_k \to x \right\} \tag{28}$$

*for some (Lebesgue) zero-measure set $\mathcal{N}_F \subset \mathbb{R}^n$ and any other zero-measure set $S \subset \mathbb{R}^n$. In particular, if $F$ is continuous at a fixed $x$, then $\mathcal{K}[F](x) = \{F(x)\}$.*

---

[11] The notions introduced here are solely needed for rigorously dealing with the singular discontinuity at the equilibrium point of the $q$-RGF and $q$-SGF flows. However, the reader can skip these definitions and still be able to intuitively follow the proofs of Theorems 1, and 2.

For instance, for the GF (1), we have $\mathcal{K}[-\nabla f](x) = \{-\nabla f(x)\}$ for every $x \in \mathbb{R}^n$, provided that $f$ is continuously differentiable. Furthermore, if $f$ is only locally Lipschitz continuous and regular (see Definition 3 of Appendix B), then $\mathcal{K}[-\nabla f](x) = -\partial f(x)$, where

$$\partial f(x) \triangleq \left\{ \lim_{k \to \infty} \nabla f(x_k) : x_k \in \mathbb{R}^n \setminus \mathcal{N}_f \text{ s.t. } x_k \to x \right\} \tag{29}$$

denotes Clarke's generalized gradient Clarke (1981) of $f$, with $\mathcal{N}_f$ denoting the zero-measure set over which $f$ is not differentiable (Rademacher's theorem). It can be established that $\partial f$ coincides with the subgradient of $f$, provided that $f$ is convex. Therefore, the GF (1) interpreted as Filippov differential inclusion may also be seen as a continuous-time variant of subgradient descent methods.

## B  FINITE-TIME STABILITY OF DIFFERENTIAL INCLUSIONS

We are now ready to focus on extending some notions from traditional Lipschitz continuous systems to differential inclusions.

**Definition 2.** We say that $x^\star \in \mathbb{R}^n$ is an *equilibrium* of (27) if $x(t) \equiv x^\star$ on some small enough non-degenerate interval is a Carathéodory solution to (27). In other words, if and only if $0 \in \mathcal{F}(x^\star)$. We say that (27) is *(Lyapunov) stable* at $x^\star \in \mathbb{R}^n$ if, for every $\varepsilon > 0$, there exists some $\delta > 0$ such that, for every maximal Carathéodory solution $x(\cdot)$ of (27), we have $\|x_0 - x^\star\| < \delta \implies \|x(t) - x^\star\| < \varepsilon$ for every $t \geq 0$ in the interval where $x(\cdot)$ is defined. Note that, under Assumption 2, if (27) is stable at $x^\star$, then $x^\star$ is an equilibrium of (27) Bacciotti & Ceragioli (1999). Furthermore, we say that (27) is *(locally and strongly) asymptotically stable* at $x^\star \in \mathbb{R}^n$ if is stable at $x^\star$ and there exists some $\delta > 0$ such that, for every maximal Carathéodory solution $x : [0, \tau) \to \mathbb{R}^n$ of (27), if $\|x_0 - x^\star\| < \delta$ then $x(t) \to x^\star$ as $t \to \tau$. Finally, (27) is *(locally and strongly) finite-time stable* at $x^\star$ if it is asymptotically stable and there exists some $\delta > 0$ and $T : B_\delta(x^\star) \to [0, \infty)$ such that, for every maximal Carathéodory solution $x(\cdot)$ of (27) with $x_0 \in B_\delta(x^\star)$, we have $\lim_{t \to T(x_0)} x(t) = x^\star$.

We will now construct a Lyapunov-based criterion adapted from the literature of finite-time stability of Lipschitz continuous systems.

**Lemma 1.** *Let $\mathcal{E}(\cdot)$ be an absolutely continuous function satisfying the differential inequality*

$$\dot{\mathcal{E}}(t) \leq -c\,\mathcal{E}(t)^\alpha \tag{30}$$

*a.e. in $t \geq 0$, with $c, \mathcal{E}(0) > 0$ and $\alpha < 1$. Then, there exists some $t^\star > 0$ such that $\mathcal{E}(t) > 0$ for $t \in [0, t^\star)$ and $\mathcal{E}(t^\star) = 0$. Furthermore, $t^\star > 0$ can be bounded by*

$$t^\star \leq \frac{\mathcal{E}(0)^{1-\alpha}}{c(1-\alpha)}, \tag{31}$$

*with this bound tight whenever (30) holds with equality. In that case, but now with $\alpha \geq 1$, then $\mathcal{E}(t) > 0$ for every $t \geq 0$, with $\lim_{t \to \infty} \mathcal{E}(t) = 0$. This will be represented by $t^\star = \infty$, with $\mathcal{E}(\infty) \triangleq \lim_{t \to \infty} \mathcal{E}(t)$.*

*Proof.* Suppose that $\mathcal{E}(t) > 0$ for every $t \in [0, T]$ with $T > 0$. Let $t^\star$ be the supremum of all such $T$'s, thus satisfying $\mathcal{E}(t) > 0$ for every $t \in [0, t^\star)$. We will now investigate $\mathcal{E}(t^\star)$. First, by continuity of $\mathcal{E}$, it follows that $\mathcal{E}(t^\star) \geq 0$. Now, by rewriting

$$\dot{\mathcal{E}}(t) \leq -c\,\mathcal{E}(t)^\alpha \iff \frac{\mathrm{d}}{\mathrm{d}t}\left[\frac{\mathcal{E}(t)^{1-\alpha}}{1-\alpha}\right] \leq -c, \tag{32}$$

a.e. in $t \in [0, t^\star)$, we can thus integrate to obtain

$$\frac{\mathcal{E}(t)^{1-\alpha}}{1-\alpha} - \frac{\mathcal{E}(0)^{1-\alpha}}{1-\alpha} \leq -c\,t, \tag{33}$$

everywhere in $t \in [0, t^\star)$, which in turn leads to

$$\mathcal{E}(t) \leq [\mathcal{E}(0)^{1-\alpha} - c(1-\alpha)t]^{1/(1-\alpha)} \tag{34}$$

and

$$t \leq \frac{\mathcal{E}(0)^{1-\alpha} - \mathcal{E}(t)^{1-\alpha}}{c(1-\alpha)} \leq \frac{\mathcal{E}(0)^{1-\alpha}}{c(1-\alpha)}, \tag{35}$$

where the last inequality follows from $\mathcal{E}(t) > 0$ for every $t \in [0, t^\star)$. Taking the supremum in (35) then leads to the upper bound (31). Finally, we conclude that $\mathcal{E}(t^\star) = 0$, since $\mathcal{E}(t^\star) > 0$ is impossible given that it would mean, due to continuity of $\mathcal{E}$, that there exists some $T > t^\star$ such that $\mathcal{E}(t) > 0$ for every $t \in [0, T]$, thus contradicting the construction of $t^\star$.

Finally, notice that if $\mathcal{E}$ is such that (30) holds with equality, then (34) and the first inequality in (35) hold with equality as well. The tightness of the bound (31) thus follows immediately. Furthermore, notice that if $\alpha \geq 1$, and $\mathcal{E}$ is a tight solution to the differential inequality (30), *i.e.* $\mathcal{E}(t) = [\mathcal{E}(0)^{1-\alpha} - c(1-\alpha)t]^{1/(1-\alpha)}$, then clearly $\mathcal{E}(t) > 0$ for every $t \geq 0$ and $\mathcal{E}(t) \to 0$ as $t \to \infty$. ∎

Cortés & Bullo (2005) proposed (Proposition 2.8) a Lyapunov-based criterion to establish finite-time stability of discontinuous systems, which fundamentally coincides with our Lemma 1 for the particular choice of exponent $\alpha = 0$. Their proposition was, however, directly based on Theorem 2 of Paden & Sastry (1987). Later, Cortés (2006) proposed a second-order Lyapunov criterion, which, on the other hand, fundamentally translates to $\mathcal{E}(t) \triangleq V(x(t))$ being strongly convex. Finally, Hui et al. (2009) generalized Proposition 2.8 of Cortés & Bullo (2005) in their Corollary 3.1, to establish semistability. Indeed, these two results coincide for isolated equilibria.

We now present a novel result that generalizes the aforementioned first-order Lyapunov-based results, by exploiting our Lemma 1. More precisely, given a Laypunov candidate function $V(\cdot)$, the objective is to set $\mathcal{E}(t) \triangleq V(x(t))$, and we aim to check that the conditions of Lemma 1 hold. To do this, and assuming $V$ to be locally Lipschitz continuous, we first borrow and adapt from Bacciotti & Ceragioli (1999) the definition of *set-valued time derivative* of $V : \mathcal{D} \to \mathbb{R}$ w.r.t. the differential inclusion (27), given by

$$\dot{V}(x) \triangleq \{a \in \mathbb{R} : \exists v \in \mathcal{F}(x) \text{ s.t. } a = p \cdot v, \forall p \in \partial V(x)\}, \tag{36}$$

for each $x \in \mathcal{D}$. Notice that, under Assumption 2 for Filippov differential inclusions $\mathcal{F} = \mathcal{K}[F]$, the set-valued time derivative of $V$ thus coincides with with the set-valued Lie derivative $\mathcal{L}_F V(\cdot)$. Indeed, more generally $\dot{V}$ could be seen as a set-valued Lie derivative $\mathcal{L}_{\mathcal{F}} V$ w.r.t. the set-valued map $\mathcal{F}$.

**Definition 3.** $V(\cdot)$ is said to be *regular* if every directional derivative, given by

$$V'(x; v) \triangleq \lim_{h \to 0} \frac{V(x + h\,v) - V(x)}{h}, \tag{37}$$

exists and is equal to

$$V^\circ(x; v) \triangleq \limsup_{x' \to x \; h \to 0^+} \frac{V(x' + h\,v) - V(x')}{h}, \tag{38}$$

known as *Clarke's upper generalized derivative* Clarke (1981).

In practice, regularity is a fairly mild and easy to guarantee condition. For instance, it would suffice that $V$ is convex or continuously differentiable to ensure that it is Lipschitz and regular.

**Assumption 3.** $V : \mathcal{D} \to \mathbb{R}$ is locally Lipscthiz continuous and regular, with $\mathcal{D} \subseteq \mathbb{R}^n$ open.

Under Assumption 3, Clarke's generalized gradient

$$\partial V(x) \triangleq \{p \in \mathbb{R}^n : V^\circ(x; v) \geq p \cdot v, \forall v \in \mathbb{R}^n\} \tag{39}$$

is non-empty for every $x \in \mathcal{D}$, and is also given by

$$\partial V(x) = \left\{ \lim_{k \to \infty} \nabla V(x_k) : x_k \in \mathbb{R}^n \setminus \mathcal{N}_V \text{ s.t. } x_k \to x \right\}, \tag{40}$$

where $\mathcal{N}_V$ denotes the set of points in $\mathcal{D} \subseteq \mathbb{R}^n$ where $V$ is not differentiable (Rademacher's theorem) Clarke (1981).

Through the following lemma (Lemma 2), we can formally establish the correspondence between the set-valued time-derivative of $V$ and the derivative of the energy function $\mathcal{E}(t) \triangleq V(x(t))$ associated with an arbitrary Carathéodory solution $x(\cdot)$ to the differential inclusion (27).

**Lemma 2** (Lemma 1 of Bacciotti & Ceragioli (1999)). *Under Assumption 3, given any Carathéodory solution $x : [0, \tau) \to \mathbb{R}^n$ to (27), then $\mathcal{E}(t) \triangleq V(x(t))$ is absolutely continuous and $\dot{\mathcal{E}}(t) = \frac{\mathrm{d}}{\mathrm{d}t} V(x(t)) \in \dot{V}(x(t))$ a.e. in $t \in [0, \tau)$.*

We are now ready to state and prove our Lyapunov-based sufficient condition for finite-time stability of differential inclusions.

**Theorem 3.** *Suppose that Assumptions 2 and 3 hold for some set-valued map $\mathcal{F} : \mathbb{R}^n \rightrightarrows \mathbb{R}^n$ and some function $V : \mathcal{D} \to \mathbb{R}$, where $\mathcal{D} \subseteq \mathbb{R}^n$ is an open and positively invariant neighborhood of a point $x^\star \in \mathbb{R}^n$. Suppose that $V$ is positive definite w.r.t. $x^\star$ and that there exist constants $c > 0$ and $\alpha < 1$ such that*

$$\sup \dot{V}(x) \leq -c\, V(x)^\alpha \tag{41}$$

*a.e. in $x \in \mathcal{D}$. Then, (27) is finite-time stable at $x^\star$, with settling time upper bounded by*

$$t^\star \leq \frac{V(x_0)^{1-\alpha}}{c(1-\alpha)}, \tag{42}$$

*where $x(0) = x_0$. In particular, any Carathéodory solution $x(\cdot)$ with $x(0) = x_0 \in \mathcal{D}$ will converge in finite time to $x^\star$ under the upper bound (42). Furthermore, if $\mathcal{D} = \mathbb{R}^n$, then (27) is globally finite-time stable. Finally, if $\dot{V}(x)$ is a singleton a.e. in $x \in \mathcal{D}$ and (41) holds with equality, then the bound (42) is tight.*

*Proof.* Note that, by Proposition 1 of Bacciotti & Ceragioli (1999), we know that (27) is Lyapunov stable at $x^\star$. All that remains to be shown is local convergence towards $x^\star$ (which must be an equilibrium) in finite time. Indeed, given any maximal solution $x : [0, t^\star) \to \mathbb{R}^n$ to (27) with $x(0) = x_0 \neq x^\star$, we know by Lemma 2, that $\mathcal{E}(t) = V(x(t))$ is absolutely continuous with $\dot{\mathcal{E}}(t) \in \dot{V}(x(t))$ a.e. in $t \in [0, t^\star)$. Therefore, we have

$$\dot{\mathcal{E}}(t) \leq \sup \dot{V}(x(t)) \leq -c\, V(x(t))^\alpha = -c\, \mathcal{E}(t)^\alpha \tag{43}$$

a.e. in $t \in [0, t^\star)$. Since $\mathcal{E}(0) = V(x_0) > 0$, given that $x_0 \neq x^\star$, the result then follows by invoking Lemma 1 and noting that $\mathcal{E}(t^\star) = 0 \iff V(t^\star, x(t^\star)) = 0 \iff x(t^\star) = x^\star$. ∎

Finite-time stability still follows without Assumption 2, provided that $x^\star$ is an equilibrium of (27). In practical terms, this means that trajectories starting arbitrarily close to $x^\star$ may not actually exist, but nevertheless there exists a neighborhood $\mathcal{D}$ of $x^\star$ over which, any trajectory $x(\cdot)$ that indeed exists and starts at $x(0) = x_0 \in \mathcal{D}$ must converge in finite time to $x^\star$, with settling time upper bounded by $T(x_0)$ (the bound still tight in the case that (41) holds with equality).

## C    PROOF OF THEOREM 1

Let us focus on the $q$-RGF (2) (the case of $q$-SGF (3) follows exactly the same steps) with the candidate Lyapunov function $V \triangleq f - f^\star$. Clearly, $V$ is Lipschitz continuous and regular (given that it is continuously differentiable). Furthermore, $V$ is positive definite w.r.t. $x^\star$.

Notice that, due to the dominated gradient assumption, $x^\star$ must be an isolated stationary point of $f$. To see this, notice that, if $x^\star$ were not an isolated stationary point, then there would have to exist some $\tilde{x}^\star$ sufficiently near $x^\star$ such that $\tilde{x}^\star$ is both a stationary point of $f$, and satisfies $f(\tilde{x}^\star) > f^\star$, since $x^\star$ is a strict local minimizer of $f$. But then, we would have

$$0 = \frac{p-1}{p} \|\nabla f(\tilde{x}^\star)\|^{\frac{p}{p-1}} \geq \mu^{\frac{1}{p-1}} (f(\tilde{x}^\star) - f^\star) > 0, \tag{44}$$

and subsequently $0 > 0$, which is absurd.

Therefore, $F(x) \triangleq -c\nabla f(x)/\|\nabla f(x)\|^{\frac{q-2}{q-1}}$ is continuous for every $x \in \mathcal{D} \setminus \{x^\star\}$, for some small enough open neighborhood $\mathcal{D}$ of $x^\star$. Let us assume that $\mathcal{D}$ is positively invariant w.r.t. (2), which can be achieved, for instance, by replacing $\mathcal{D}$ with its intersection with some small enough strict sublevel set of $f$. Notice that $\|F(x)\| = c\|\nabla f(x)\|^{\frac{1}{q-1}}$ with $q \in (p, \infty] \subset (1, \infty]$, i.e., $\frac{1}{q-1} \in [0, \infty)$. If $q = \infty$, which results in the normalized gradient flow $\dot{x} = -\frac{\nabla f(x)}{\|\nabla f(x)\|}$ proposed by Cortés (2006),

then $\|F(x)\| = c > 0$. We can thus show that $F(x)$ is discontinuous at $x = 0$ for $q = \infty$. On the other hand, if $q \in (p, \infty) \subset (1, \infty)$, then we have $\|F(x)\| \to 0$ as $x \to x^\star$, and thus $F(x)$ is continuous (but not Lipschitz) at $x = x^\star$. Regardless, we may freely focus exclusively on $\mathcal{D} \setminus \{x^\star\}$ since $\{x^\star\}$ is obviously a zero-measure set.

Let $\mathcal{F} \triangleq \mathcal{K}[F]$. We thus have, for each $x \in \mathcal{D} \setminus \{x^\star\}$,

$$\sup \dot{V}(x) = \sup \left\{ a \in \mathbb{R} : \exists v \in \mathcal{F}(x) \text{ s.t. } a = p \cdot v, \forall p \in \partial V(x) \right\} \tag{45a}$$

$$= \sup \left\{ \nabla V(x) \cdot v : v \in \mathcal{F}(x) \right\} \tag{45b}$$

$$= \nabla V(x) \cdot F(x) \tag{45c}$$

$$= -c \|\nabla f(x)\|^{2 - \frac{q-2}{q-1}} \tag{45d}$$

$$= -c \|\nabla f(x)\|^{\frac{1}{\theta'}} \tag{45e}$$

$$\leq -c [C(f(x) - f^\star)^\theta]^{\frac{1}{\theta'}} \tag{45f}$$

$$= -c C^{\frac{1}{\theta'}} V(x)^{\frac{\theta}{\theta'}}. \tag{45g}$$

Since $\frac{\theta}{\theta'} < 1$, given that $s > 1 \mapsto \frac{s-1}{s}$ is strictly increasing, then the conditions of Theorem 3 are satisfied. In particular, we have finite-time stability at $x^\star$ with a settling time $t^\star$ upper bounded by

$$t^\star \leq \frac{(f(x_0) - f^\star)^{1 - \frac{\theta}{\theta'}}}{c \, C^{\frac{1}{\theta'}} \left(1 - \frac{\theta}{\theta'}\right)} \leq \frac{(\|\nabla f(x_0)\|/C)^{\frac{1}{\theta}\left(1 - \frac{\theta}{\theta'}\right)}}{c \, C^{\frac{1}{\theta'}} \left(1 - \frac{\theta}{\theta'}\right)} = \frac{\|\nabla f(x_0)\|^{\frac{1}{\theta} - \frac{1}{\theta'}}}{c C^{\frac{1}{\theta}} \left(1 - \frac{\theta}{\theta'}\right)} \tag{46}$$

for each $x_0 \in \mathcal{D}$, which completes the proof.

## D  PROOF OF THEOREM 2

To prove Theorem 2, we borrow some tools and results from hybrid control systems theory. Hybrid control systems are characterized by continuous flows with discrete jumps between the continuous flows. They are often modeled by differential inclusions added to discrete mappings to model the jumps between the differential inclusions. We see the case of the optimization flows proposed here as a simple case of a hybrid systems with one differential inclusion, with a possible jump or discontinuity at the optimum. Based on this, we will use the tools and results of Sanfelice & Teel (2010), which study how a certain class of hybrid systems behave after discretization with a certain class of discretization algorithms. In other words, Sanfelice & Teel (2010) quantifies, under some conditions, how close are the solutions of the discretized hybrid dynamical system to the solutions of the original hybrid system.

In this section we will denote the differential inclusion of the continuous optimization flow by $\mathcal{F} : \mathbb{R}^n \rightrightarrows \mathbb{R}^n$, and its discretization in time by $\mathcal{F}_{\mathrm{d}} : \mathbb{R}^n \rightrightarrows \mathbb{R}^n$. We first recall a definition, which we will adapt from the general case of jumps between multiple differential inclusions (Definition 3.2, Sanfelice & Teel (2010)) to our case of one differential inclusion or flow.

**Definition 4.** *($(T, \epsilon)$-closeness).* Given $T > 0$, $\epsilon > 0$, $\eta > 0$, two solutions $x_t : [0, T] \to \mathbb{R}^n$, and $x_k : \{0, 1, 2, ...\} \to \mathbb{R}^n$ are $(T, \epsilon)$-close if:
(a) for all $t \leq T$ there exists $k \in \{1, 2, ...\}$ such that $|t - k\eta| < \epsilon$, and $\|x_t(t) - x_k(k)\| < \epsilon$,
(b) for all $k \in \{1, 2, ...\}$ there exists $t \leq T$ such that $|t - k\eta| < \epsilon$, and $\|x_t(t) - x_k(k)\| < \epsilon$.

Next, we will recall Theorem 5.2 in Sanfelice & Teel (2010), while adapting it to our special case of a hybrid system with one differential inclusion[12].

**Theorem 4.** *(Closeness of continuous and discrete solutions on compact domains) Consider the differential inclusion*

$$\dot{X}(t) \in \mathcal{F}(X(t)), \tag{47}$$

---

[12] A set-valued mapping $\mathcal{F} : \mathbb{R}^n \rightrightarrows \mathbb{R}^n$ is *outer semicontinuous* if for each sequence $\{x_i\}_{i=1}^\infty$ converging to a point $x \in \mathbb{R}^n$ and each sequence $y_i \in \mathcal{F}(x_i)$ converging to a point $y$, it holds that $y \in \mathcal{F}(x)$. It is *locally bounded* if, for each $x \in \mathbb{R}^n$, there exists compact sets $K, K' \subset \mathbb{R}^n$ such that $x \in K$ and $\mathcal{F}(K) \triangleq \cup_{x \in K} \mathcal{F}(x) \subset K'$. In what follows, we use the following notations: Given a set $A$, $\mathrm{con} A$ denotes the convex hull, and $\mathbb{B}$ denotes the closed unit ball in a Euclidean space.

*for a given set-valued mapping $\mathcal{F} : \mathbb{R}^m \rightrightarrows \mathbb{R}^m$ assumed to be outer semicontinuous, locally bounded, nonempty, and with convex values for every $x \in \mathcal{C}$, for some closed set $\mathcal{C} \subseteq \mathbb{R}^m$. Consider the discrete-time system represented by the flow $\mathcal{F}_d : \mathbb{R}^n \rightrightarrows \mathbb{R}^n$, such that, for each compact set $K \subset \mathbb{R}^n$, there exists $\rho \in \mathcal{K}_\infty$, and $\eta^\star > 0$ such that for each $x \in K$ and each $\eta \in (0, \eta^\star]$,*

$$\mathcal{F}_d(x) \subset x + \eta \operatorname{con}\mathcal{F}(x + \rho(\eta)\mathbb{B}) + \eta\rho(\eta)\mathbb{B}. \tag{48}$$

*Then, for every compact set $K \subset \mathbb{R}^n$, every $\epsilon > 0$, and every time horizon $T \in \mathbb{R}_{\geq 0}$ there exists $\eta^\star > 0$ such that: for any $\eta \in (0, \eta^\star]$ and any discrete solution $x_k$ with $x_k(0) \in K + \delta\mathbb{B}$, $\delta > 0$, there exists a continuous solution $x_t$ with $x_t(0) \in K$ such that $x_k$ and $x_t$ are $(T, \epsilon)$-close.*

To prove Theorem 2 we will use the results of Theorem 4, where we will have to check that condition (48) is satisfied for the three proposed discretizations.

We are now ready to prove Theorem 2. First, note that outer semicontinuity follows from the upper semicontinuity and the closedness of the Filippov differential inclusion map. Furthermore, local boundedness follows from continuity everywhere outside stationary points, which are isolated.

Now, let us examine their discretization by the three proposed algorithms:

## FORWARD-EULER DISCRETIZATION

The mapping $\mathcal{F}_d$ in this case is a singleton, given by

$$\mathcal{F}_d(x) \triangleq x + \eta F(x), \tag{49}$$

where $\eta > 0$, which clearly satisfies condition (48).

## RUNGE-KUTTA DISCRETIZATION

Once again, the mapping $\mathcal{F}_d$ is singleton, this time given by

$$\mathcal{F}_d(x) \triangleq x + \eta \sum_{i=1}^{K} \alpha_i \mathcal{F}(y^i) \tag{50a}$$

$$y^i = x + \eta \sum_{j=1}^{i-1} \beta_j \mathcal{F}(y^j), \tag{50b}$$

where $\eta, \alpha_1, \ldots, \alpha_K, \beta_1, \ldots, \beta_{K-1} > 0$ are such that $\sum_{i=1}^{K} \alpha_i = 1$.

By equation (50b) one can establish a function $\rho \in \mathcal{K}_\infty$ such that for each $x_k \in K \subset \mathbb{R}^n$, for each $\eta > 0$, $y_k^i \in x_k + \eta\rho(\eta)\mathbb{B}$. Next, by equation (50a) together with the condition $\sum_{i=1}^{i=K} \alpha_i = 1$, one can write that for any $x_k \in K$ and $\eta > 0$, $\mathcal{F}_d(x_k) \subset x_k + \eta\operatorname{con}\mathcal{F}(y_k^i) \subset x_k + \eta\operatorname{con}\mathcal{F}(x_k + \eta\rho(\eta)\mathbb{B}) + \eta\rho(\eta)\mathbb{B}$.

## NESTEROV-LIKE DISCRETIZATION

In this case the discrete-time flow $\mathcal{F}_d$ is defined as

$$\mathcal{F}_d(x_k) = x_k + \eta\mathcal{F}(x_k + \mu y_k) + \mu y_k \tag{51a}$$

$$y_k = x_k - x_{k-1}. \tag{51b}$$

In this case, to take into account the integral effect of the Nesterov-like discretization, let us extend the continuous-time flow as

$$\dot{\tilde{x}}_t = \begin{pmatrix} \dot{x}_t \\ \frac{d}{dt} \frac{\int_t^{t+\eta} x_t(s)ds}{\eta} \end{pmatrix} = \begin{pmatrix} \mathcal{F}(x_t) \\ \frac{x_{t+\eta} - x_t}{\eta} \end{pmatrix}, \tag{52}$$

we then compare the solution of the extended continuous-time system (52) with the extended discrete-time system

$$\tilde{x}_{k+1} = \begin{pmatrix} x_{k+1} \\ x_{k+1} - x_k \end{pmatrix} = \begin{pmatrix} x_k + \eta\mathcal{F}(x_k(1+\mu) + x_{k-1}) + \mu(x_k - x_{k-1}) \\ \eta\mathcal{F}(x_k(1+\mu) + x_{k-1}) + \mu(x_k - x_{k-1}) \end{pmatrix}, \tag{53}$$

which could be rearranged as

$$\mathcal{F}_{\mathrm{d}}(\tilde{x}_k) = \tilde{x}_{k+1} = \tilde{x}_k + M\tilde{x}_k + \eta\tilde{\mathcal{F}}_t(\tilde{x}_k), \tag{54}$$

where $M = \begin{bmatrix} 0 & \mu \\ 0 & -1+\mu \end{bmatrix}$, and $\tilde{\mathcal{F}}_t(\tilde{x}_k) = \begin{bmatrix} \mathcal{F}(\tilde{x}_k) \\ \mathcal{F}(\tilde{x}_k) \end{bmatrix}$, which shows that for any $\tilde{x}_k \in K \subset \mathbb{R}^{2n}$, and $\eta > 0$, we have (using similar recursive reasoning as above) that $\mathcal{F}_{\mathrm{d}}(\tilde{x}_k) \subset \tilde{x}_k + \eta\mathrm{con}\tilde{\mathcal{F}}_t(\tilde{x}_k + \eta\rho(\eta)\mathbb{B}) + \eta\rho(\eta)\mathbb{B}$. Then, using Theorem 4 we conclude about the $(T, \epsilon)$-closeness between the continuous-time solutions of the flows $\mathcal{F} : q$-RGF (2), $q$-SGF (3), and the discrete-time solutions of their respective discretization by any of the three discretization methods. Furthermore, for the Nesterov-like discretization, we can also conclude about the $(T, \epsilon)-$ closeness of the integral of the continuous-time solutions $\tilde{x}_t(2)$ and its discretization $\tilde{x}_k(2)$.

Finally, using the Lyapunov function $V = f - f^\star$ as defined in the proof of Theorem 1, together with inequalities (45g), (34), and a local Lipschitz bound on $f$, one can derive the convergence bound given by (23), as follows:

$$\begin{aligned} \|f(x_k) - f(x^\star) - (f(x_t) - f(x^\star))\| &= \|f(x_k) - f(x_t)\| \le \tilde{\epsilon} = L_f\epsilon, \; L_f > 0, \; \epsilon > 0, \\ \|f(x_k) - f(x^\star)\| - \|(f(x_t) - f(x^\star))\| &\le \|f(x_k) - f(x_t)\| \le \tilde{\epsilon}, \\ \|f(x_k) - f(x^\star)\| &\le \tilde{\epsilon} + \|f(x_t) - f(x^\star)\|, \\ \|f(x_k) - f(x^\star)\| &\le \tilde{\epsilon} + [(f(x_0) - f(x^\star))^{(1-\alpha)} - \tilde{c}(1-\alpha)\eta k]^{1/(1-\alpha)}, \text{for } k \le k^\star, \end{aligned}$$

where $\alpha = \frac{\theta}{\theta'}$, $\theta = \frac{p-1}{p}$, $\theta' = \frac{q-1}{q}$, $\tilde{c} = c\left(\left(\frac{p}{p-1}\right)^{\frac{p-1}{p}}\mu^{\frac{1}{p}}\right)^{\frac{1}{\theta'}}$, $k^\star = \frac{(f(x_0)-f(x^\star))^{(1-\alpha)}}{\tilde{c}(1-\alpha)\eta}$.

# E  ADDITIONAL DETAILS AND NUMERICAL RESULTS

In this section, we will expand upon the numerical results experiments discussed in the paper. In particular, we report more details on the hyper-parameters values used[13] for the numerical tests, and report some results for the MNIST experiments.

## E.1  HYPER PARAMETERS VALUES USED IN THE TESTS OF SECTION 4.1

- GD fixed step size: $\eta = 10^{-3}$
- RGF Euler disc. w/fixed step size: $q = 2.2$, $\eta = 10^{-3}$
- RGF Euler disc. w/fixed step size: $q = 3$, $\eta = 10^{-2}$
- RGF Euler disc. w/fixed step size: $q = 6$, $\eta = 10^{-2}$
- RGF Euler disc. w/fixed step size: $q = 10$, $\eta = 10^{-2}$
- GD Nesterov acceleration fixed step size: $\eta = 10^{-4}; \mu = 0.9$
- SGF Nesterov-like disc. w/fixed step size: $q = 2.2$, $\eta = 10^{-4}$, $\mu = 0.9$
- SGF Nesterov-like disc. w/fixed step size: $q = 3$, $\eta = 10^{-3}$, $\mu = 0.9$
- SGF Nesterov-like disc. w/fixed step size: $q = 6$, $\eta = 10^{-3}$, $\mu = 0.9$
- SGF Nesterov-like disc. w/fixed step size: $q = 10$, $\eta = 10^{-2}$, $\mu = 0.09$
- RGF Runge Kutta disc. w/fixed step size: $q = 2.2$, $K = 2$, $\eta = 10^{-2}$, $\beta_1 = 0.09$, $\alpha_1 = \alpha_2 = 0.5$
- RGF Runge Kutta disc. w/fixed step size: $q = 3$, $K = 2$, $\eta = 10^{-2}$, $\beta_1 = 0.09$, $\alpha_1 = \alpha_2 = 0.5$
- RGF Runge Kutta disc. w/fixed step size: $q = 6$, $K = 2$, $\eta = 10^{-2}$, $\beta_1 = 0.09$, $\alpha_1 = \alpha_2 = 0.5$
- RGF Runge Kutta disc. w/fixed step size: $q = 10$, $K = 2$, $\eta = 10^{-2}$, $\beta_1 = 0.09$, $\alpha_1 = \alpha_2 = 0.5$

---

[13]In all the tests, for $q$-RGF and $q$-SGF $c = 1$ unless otherwise stated.

### E.2 Hyper parameters values used in the tests of Section 4.2

Note that the description of the coefficients for each of the prior art methods can be found in: https://pytorch.org/docs/stable/optim.html.

- GD: $\eta = 4.10^{-2}$, $\mu = 0.9$, Nesterov=True
- RGF: $\eta = 4.10^{-2}$, $\mu = 0.9$
- SGF: $\eta = 4.10^{-3}$, $\mu = 0.9$
- ADAM: $\eta = 8.10^{-4}$ (remaining coefficients=nominal values)
- RMS: $\eta = 10^{-3}$ (remaining coefficients=nominal values)
- ADAGRAD: $\eta = 10^{-3}$ (remaining coefficients=nominal values)
- ADADELTA: $\eta = 4.10^{-2}$, $\rho = 0.9$, $\epsilon = 10^{-6}$, weight decay $= 0$

### E.3 More tests on the Rosenbrock function

Due to space limitations, we have decided to report in the main paper only one test for $q$-RGF with Euler discretization, one test for $q$-SGF with Nesterov-like discretization, and one test for $q$-RGF with Runge Kutta discretization. For the sake of completeness we report here the remaining tests for each algorithm. One can observe similar qualitative behavior in Figure 6 as the one noticed in the results of Section 4.5.

The step-size and other hyper-parameters for each test are given below:

- GD fixed step size: $\eta = 10^{-3}$
- SGF Euler disc. w/fixed step size: $q = 2.1$, $\eta = 10^{-3}$
- SGF Euler disc. w/fixed step size: $q = 2.5$, $\eta = 10^{-3}$
- SGF Euler disc. w/fixed step size: $q = 2.8$, $\eta = 10^{-3}$
- SGF Euler disc. w/fixed step size: $q = 100$, $\eta = 10^{-2}$
- GD Nesterov acceleration fixed step size: $\eta = 10^{-4}$; $\mu = 0.9$
- RGF Nesterov-like disc. w/fixed step size: $q = 2.2$, $\eta = 10^{-4}$, $\mu = 0.9$
- RGF Nesterov-like disc. w/fixed step size: $q = 3$, $\eta = 10^{-3}$, $\mu = 0.9$
- RGF Nesterov-like disc. w/fixed step size: $q = 6$, $\eta = 10^{-3}$, $\mu = 0.9$
- RGF Nesterov-like disc. w/fixed step size: $q = 10$, $\eta = 10^{-3}$, $\mu = 0.9$
- SGF Runge Kutta disc. w/fixed step size: $q = 2.2$, $K = 2$, $\eta = 10^{-3}$, $\beta_1 = 0.09$, $\alpha_1 = \alpha_2 = 0.5$
- SGF Runge Kutta disc. w/fixed step size: $q = 3$, $K = 2$, $\eta = 10^{-2}$, $\beta_1 = 0.9$, $\alpha_1 = \alpha_2 = 0.5$
- SGF Runge Kutta disc. w/fixed step size: $q = 6$, $K = 2$, $\eta = 10^{-2}$, $\beta_1 = 0.09$, $\alpha_1 = \alpha_2 = 0.5$
- SGF Runge Kutta disc. w/fixed step size: $q = 10$, $K = 2$, $\eta = 10^{-2}$, $\beta_1 = 0.09$, $\alpha_1 = \alpha_2 = 0.5$

**Remark 2. Choice of $q$:** The settling time upper bound (15) decreases as $q \to \infty$, which appears to lead to faster convergence when discretized. On the other hand, the larger $q$ is, the stiffer the ODE, so more prone to numerical instability, so $q$ cannot be too large. Therefore, assuming $p$ to be not too large, it appears that $q \in (p, p + \delta]$ works best, with $\delta > 0$ as small as needed to avoid numerical issues. For instance, if we know the cost function to be strongly convex (locally), then we search for $q$ slightly larger than $p = 2$ at first, but continue to increase until performance deteriorates. If, on the other hand, we don't know the order $p > 1$, then it's currently unclear how to choose $q$. We will investigate this further in future work. Furthermore, there is evidence that gradient dominance does hold locally in many deep learning contexts (Zhou and Liang, 2017, https://arxiv.org/abs/1710.06910). Indeed, since convexity readily leads to gradient dominance of order $p = \infty$, it suffices that a slightly stronger form of it holds (but weaker than strong convexity), in order to have $p < \infty$, and thus for us to be able to choose $q > p$.

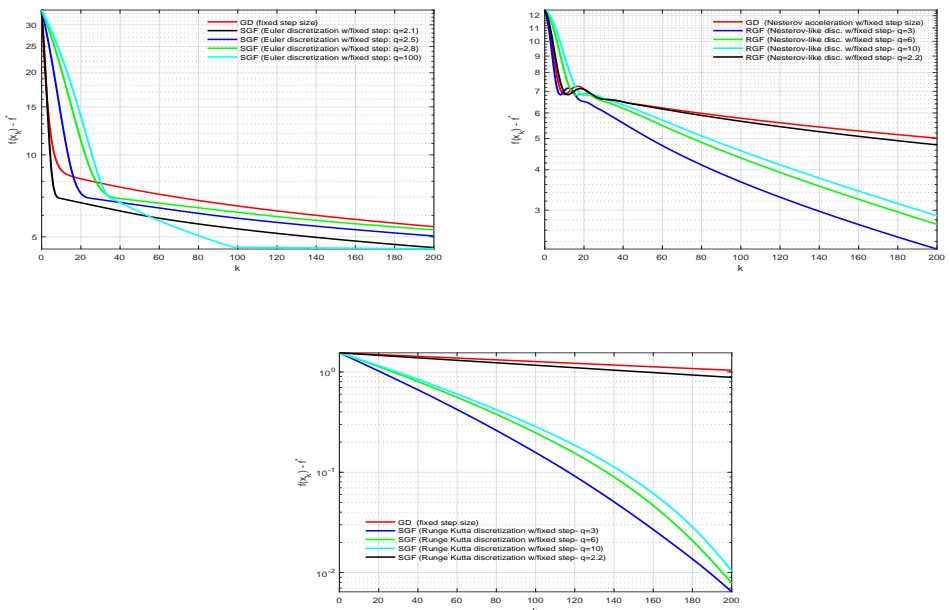

Figure 6: Example of the proposed discretization algorithms of finite-time $q$-RGF and $q$-SGF

### E.4   EXPERIMENT 3: MNIST CNN MODEL

In this experiment we optimize the CNN network described using a Pytorch code sequence in **MODEL 1** with a negative log likelihood loss, on the MNIST dataset.

We use 10 epochs of training, with 60 batches of 1000 images for training, and 10 batches of 1000 images for testing. We tested Algorithm 1 RGF ($c = 1$, $q = 3$, $\eta = 0.06$, $\mu = 0.9$), and Algorithm 2 SGF ($c = 0.001$, $q = 2.1$, $\eta = 0.06$, $\mu = 0.9$) against Nesterov's accelerated gradient descent (GD) ($\eta = 0.06$, $\mu = 0.9$, Nesterov=True), and Adam ($\eta = 0.004$, remaining coefficients=nominal values of Torch.Optim). We also tested other algorithms such as RMSprop, AdaGrad, and AdaDelta, but since their convergence performance, as well as, test performance were not very competitive w.r.t. GD, on this experiment, we decided not to report them here, to avoid overloading the graphs. In Figures 7, 8 we can see the training loss over the training iterations, where we see that GD, RGF and SGF perform better than Adam in terms of convergence speed ($20\ sec$ lead in average), and in terms of test performance $98\%$ for Adam, $98\%$ for GD, and $99\%$ for both RGF and SGF. The RGF and SGF perform slightly better than GD in terms of convergence speed. The gain is relatively small ($5\ sec$ to $10\ sec$ in average) which is expected in such a small DNN network (please refer to VGG16-CIFAR10, and VGG16-SVHN test results for larger computation-time gains). We also notice, in Figure 7 that all algorithms behave well in terms of avoiding overfitting the model.

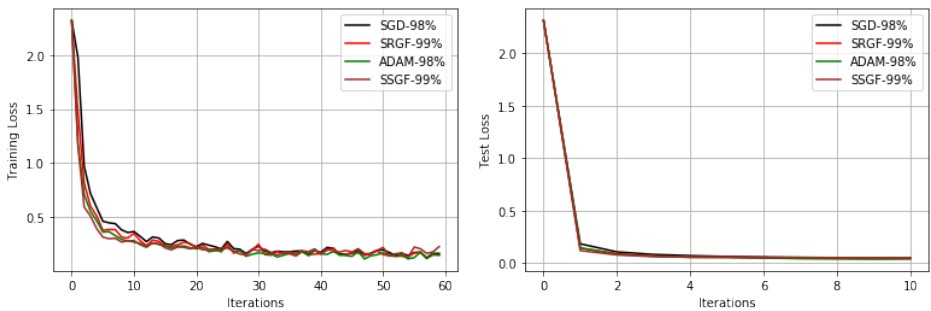

Figure 7: Losses for several optimization algorithms- CNN- MNIST: Train loss (left), test loss (right)

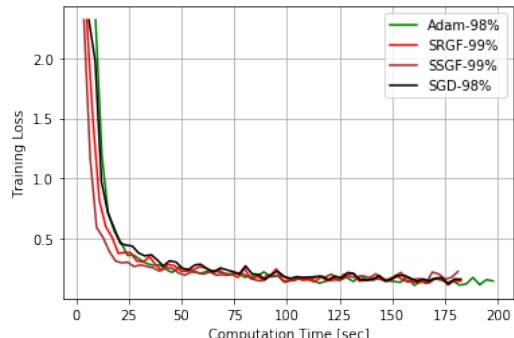

Figure 8: Training loss vs. computation time for several optimization algorithms- MNIST

**MODEL 1:** MNIST-CNN

```
class Net(MNIST-CNN)
  def.init.(self):
  super(Net, self).init()
  self.conv1 = nn.Conv2d(1, 10, kernel.size=5)
  self.conv2 = nn.Conv2d(10, 20, kernel.size=5)
  self.conv2.drop = nn.Dropout2d()
  self.fc1 = nn.Linear(320, 50)
  self.fc2 = nn.Linear(50, 10)

def forward(self, x):
  x = F.relu(F.max.pool2d(self.conv1(x), 2))
  x = F.relu(F.max.pool2d(self.conv2.drop(self.conv2(x)), 2))
  x = x.view(-1, 320)
  x = F.relu(self.fc1(x))
  x = F.dropout(x, training=self.training)
  x = self.fc2(x)
return F.log.softmax(x)
```

## E.5    EXPERIMENT 4: EULER DISCRETIZATION ON SVHN DATASET

In the main paper, due to space limitation, we decided to report the results of the Nesterov-like discretization only, which also seems like a fair comparison since our Nesterov-like discretization of $q$-RGF and $q$-SGF flows can be compared against Nesterov implementation of GD algorithm. However, we also wanted to test the performance of the simple Euler discretization of the proposed flows against the simple GD algorithm, to do so we run some extra tests on the SVHN dataset. These results are presented below.

We tested the proposed Euler algorithms to train the same VGG16 CNN model with cross entropy loss. We divided the dataset into a training set of 70 batches with 1000 images each, and a test set of 10 of 1000 images each, and ran 20 epochs of training over all the training batches. We tested the Euler discretization of $q$-RGF ($c = 1$, $q = 2.1$, $\eta = 0.04$, $\mu = 0.9$), and the Euler discretization of $q$-SGF ($c = 10^{-3}$, $q = 2.1$, $\eta = 0.04$, $\mu = 0.9$) against gradient descent (GD) and Adam (same optimal tuning as in Section 4.2). All algorithms have been implemented in their stochastic version.

In Figures 9 , 10 we can see that both algorithms, Euler $q$-RGF and Euler $q$-SGF, converge faster ($40\ min$ lead in average) than SGD and Adam for these tests, and reach an overall better performance on the test-set.

## E.6    EXPERIMENT 5: RUNGE-KUTTA DISCRETIZATION ON SVHN DATASET

Finally, for compleetness, we also wanted to test the performance of the Runge-Kutta discretization of the proposed flows against SGD, to do so we run some extra tests on the SVHN dataset. These results are presented below.

We tested the proposed Runge-Kutta algorithms to train the same VGG16 CNN model with cross entropy loss. We divided the dataset into a training set of 70 batches with 1000 images each, and a test set of 10 of 1000 images each, and ran 20 epochs of training over all the training batches. We tested the Runge-Kutta discretization of $q$-RGF ($c = 1$, $q = 2.1$, $K = 2$, $\eta = 10^{-2}$, $\beta_1 = 10^{-2}$, $\alpha_1 = \alpha_2 = 0.5$), and the Runge-Kutta discretization of $q$-SGF ($c = 10^{-3}$, $q = 2.1$, $K = 2$, $\eta = 10^{-2}$, $\beta_1 = 10^{-2}$, $\alpha_1 = \alpha_2 = 0.5$) against

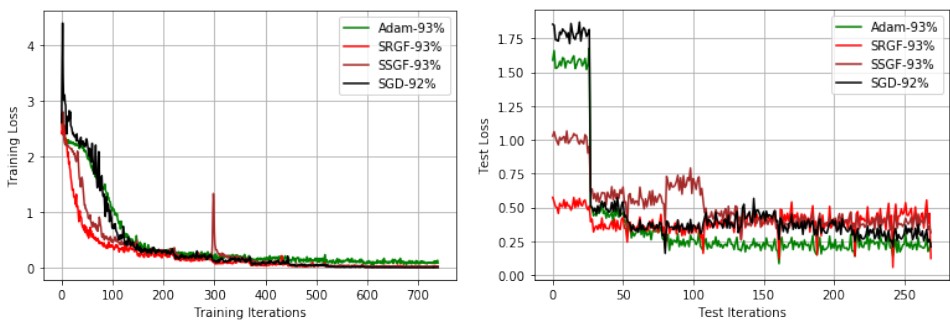

Figure 9: Losses for several optimization algorithms- SVHN: Train loss (left), test loss (right)

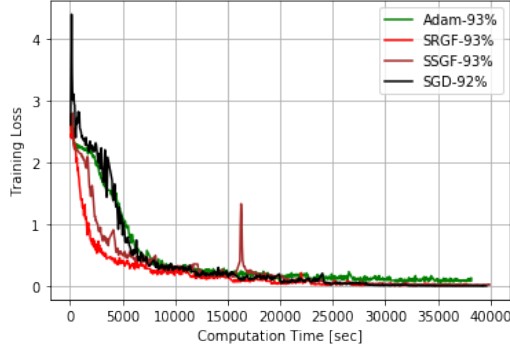

Figure 10: Training loss vs. computation time for several optimization algorithms- VGG-16- SVHN

gradient descent (GD) and Adam (same optimal tuning as in Section 4.2). All algorithms have been implemented in their stochastic version.

In Figures 11, 12 we can see that both algorithms, Runge-Kutta $q$-RGF and Runge-Kutta $q$-SGF converge faster (40 $min$ lead in average) than SGD and Adam for these tests.

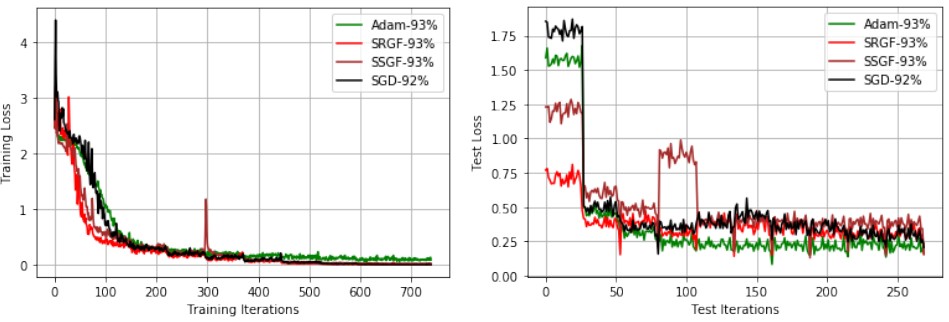

Figure 11: Losses for several optimization algorithms- SVHN: Train loss (left), test loss (right)

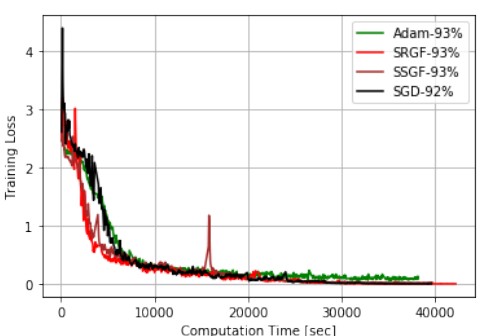

Figure 12: Training loss vs. computation time for several optimization algorithms- VGG-16- SVHN

