# OpenReview forum: "First-Order Optimization Algorithms via Discretization of Finite-Time Convergent Flows"
_ICLR.cc/2021/Conference — Reject_

### Official Review · AnonReviewer2 · 2020-10-24
**Report**

**Rating:** 6
**Confidence:** 4

**Review:**

Summary: This paper studies inertial algorithms motivated from discretization of continuous-time systems. The focus of this paper is on rescaled gradient flows and studies three different numerical discretization schemes. The performance of the thus obtained schemes is tested on standard test instances in deep learning.

Evaluation:
The paper treats a very important and interesting topic. I very much like the idea of using rescaled gradient flows for optimization problems. However, the scheme described is conceptual in nature as many unexplained hyper parameters play a key role in the optimization and no indication of how to choose these hyper parameters is given. Furthermore, I am missing a discussion with a comparison between the described method and the other first-order methods under the stated gradient-dominance condition. Furthermore, the writing is partly a bit imprecise and I am not quite sure to fully understand the main arguments used in the proof of Theorem 2. If the authors are able to explain this a bit better, then I would be in principle willing to increase my evaluation of this paper.


Pros:
+ The analysis of numerical discretization techniques of rescaled gradient flows is a topic that received a lot of attention and definitely is an important subject.
+ A complexity estimate of the numerical schemes is given.
+ Numerical results
+ Detailed technical details are provided in the appendix.

Cons:
- The meaning of Eqs. (6a),(6b) is unclear to me. The map $G$ seems to be not really playing any role in the Examples 1-3.
- The gradient dominance condition reduces in the case $p=2$ to the well-known Polyak-Lojasiewicz condition. Polyak proved already in 1963 that simple GD features linear convergence under this condition. Attouch-Bolte (2009) generalized this to significantly. These connections are not mentioned at all but are central to the approach here.
- I don't understand why the algorithms stops once and $\epsilon$-neighborhood of the solution is reached. Is this a stopping condition in the numerical approach? If so, it must be stated explicitly. Also, do you really mean $\Delta f(x^{*})=0$ for the equilibrium condition on p. 6?
- I don't understand why the connection between the numerical scheme and hybrid systems has to be made to prove that approximation property. There is a solid theory of stochastic approximation of set-valued dynamical systems due to Benaim, Hofbauer and Sorin (SIAM J. CONTROL OPTIM. 2005, Vol. 44, No. 1, pp. 328–348) where the needed tubular estimates are provided in quite some generality using the theory of perturbed solutions to differential inclusions. I belief that it is more natural to relate the approximation results to these techniques. However, I might miss this point as the authors write on p. 16 that they consider the dynamics as a hybrid system with a possible jump at the optimum. In my opinion this is an imprecise formulation and should be reconsidered.
- The methods are not really algorithms but rather conceptual computational schemes.
- I don’t understand the statement in Theorem 2 why the bound is a „weak convergence statement“. Can you explain this terminology?
- There is no discussion on the efficiency of the method although gradient dominated functions have been studied in the literature to quite some extend.

---

> ### Author Response · Authors · 2020-11-12
> **Thank you for your comments; please see our explanations below**
>
> {Q1- Evaluation: The paper treats a very important...my evaluation of this paper}:
>
> R1- Thank you for your review work. We appreciate that this reviewer is willing to give the paper a chance to be presented and discussed with our colleagues at the conference. We will try to better explain the confusing points of the paper, here and, when appropriate, in the revised version of the paper as well.
>
> {Q2- The meaning of Eqs. (6a),(6b) is unclear... in the Examples 1-3}:
>
> R2- Indeed, we only used such general map to simplify the presentation, i.e., transiting from general continuous flows as in (4), (5), to general discrete flows as in (6). We thought that the notions of state-space representation in (6) can help us put the work in the context of control-theory state representation, as used in the proof of Theorem 2, in Appendix. Besides, writing the discrete optimization steps as in (6) gave us a more general framework allowing us to simply present examples 1-3, while defining the corresponding mapping $F_d$ and $G$ for each example. Please note that these mapping are indeed defined for each example, i.e., Example 1, in the second line after equation (7); Example 2, in the second line after equation (9); and Example 3, in equation (12) and the first line after (12).
>
> {Q3-  The gradient dominance condition...approach here}:
>
> R3- We agree, we have revised the paper by citing the suggested references and adding more explanations about the connection between gradient dominance and the PL condition, introduced by Polyak in 1963. Thank you.
>
> {Q4-  I don't understand... condition on p. 6?}:
>
> R4- Here again we agree, the writing was a gross simplification. We meant to say that if we reach exactly $x^{\star}$, the algorithm reaches a stationary point, i.e., $x_{k+1}=x_{k},\;\forall k>k^{\star}$. However, since we are simply reaching an arbitrarily small $\epsilon$-neighborhood of $x^{\star}$, then the equality should be approximated $x_{k+1}\approx x_{k},\; \forall k>k^{\star}$, which means that one needs to explicitly implement a stopping condition on the closeness of $x_{k+1}$ and $x_{k}$. Indeed, using the $\Delta (.)$ operator is a clear typo, we meant that the gradient of the cost vanishes. More precisely, we will write that at the equilibrium point $x^{*}$ of the continuous flows $F(x^{\star})=0$ (in the case of differential equations), and $0\in \mathcal{F}(x^{\star})
> $. We have amended these points in the revised version of the paper, thank you.
>
> {Q5-  I don't understand why  the connection...be reconsidered.}:
>
> R5- We agree, there are maybe several ways of approaching this proof. Our background being of control theory, we choose to approach this `shadowing' proof from the perspective of hybrid control theory dealing with systems with several continuous flows and a switching rule dictating the switching between the continuous flows. In our setting, the hybrid system boils down  to one continuous flow with only one switching to a constant point (trivial flow) which occurs at the equilibrium point. Again this might be seen as heavy-handed since we only have one isolated discontinuity point, but this was more intuitive to us. We have looked at the paper suggested by the reviewer, and we agree that we could follow such theory to prove the approximation result. Furthermore, the suggested theory might (we need to look into the details of the assumptions used in Benaim et al. 2005) allow us to deal with the stochastic implementation of the proposed discretizations as well. We have added a statement about this point in the revised version of the paper, and are thankful to the reviewer for their constructive and helpful suggestions, indeed.
>
> {Q6- The methods are not really algorithms but rather conceptual computational schemes.}:
>
> R6- We agree that the methods are numerical discretizations, and have removed the word 'algorithm' from the paper.
>
> {Q7- I don’t understand the statement...Can you explain this terminology?}:
>
> R7- What we meant by weak convergence bound is that we don't have an explicit convergence rate. We used similar terminology as the one used in  Barakat and Bianchi in arXiv:1810.02263, 2019 (https://arxiv.org/abs/1810.02263), when they derive a similar approximation bound for the ADAM algorithm. However, to avoid confusion with other potential meanings of weak bounds, we removed this word from the paper.
>
> {Q8- There is no discussion on the efficiency of the method although gradient dominated functions have been studied in the literature to quite some extend.}:
>
> R8: Since we wanted to test and report the performance of the proposed methods in DNN setting, we thought that it is best to compare them against the most efficient optimization methods in this context, i.e., Nesterov's accelerated GD, Adam, and their variants. Some of them do have well studied convergence rates when dealing with gradient dominated functions, e.g., GD, and some have shown empirical efficiency when dealing with DNN optimization.

---

### Official Review · AnonReviewer1 · 2020-10-28
**Official Blind Review #1**

**Rating:** 4
**Confidence:** 4

**Review:**

The submission analyzes convergence behavior of three numerical discretizations (forward Euler, explicit Runge-Kutta, and Nesterov-based) of the q-rescaled gradient flow proposed by Wibisono et al. and a variant (q-signed gradient flow), and performs experiments to demonstrate improved convergence speed.


The paper is written clearly, and its relation with prior work is adequately addressed to the best of my knowledge.
However, I also believe that the paper has a few serious drawbacks, which makes it a stretch to include it in the conference in the current form. In general, I feel the submission has the potential in becoming a good paper after some non-trivial updates.


The first issue is with regards to the motivation. After reading the paper, I don’t fully see the benefit that finite-time continuous time convergence provides. What’s more is that the submission does not well motivate the reason to use the particular flows and the techniques of analyses which rely on differential inclusion.


The second issue I’d like to raise is perhaps more important and really is the deal breaker in my opinion. It seems the only new theoretical contribution (i.e. theorem 2) does not provide a converging bound with which a convergence rate can be derived. This is demonstrated by the fact that the authors acknowledge in the theorem that epsilon is chosen, i.e. it cannot be arbitrary. It is also unclear how small a step size eta is required for this bound to hold. Skimming the proof in the appendix, it is clear that the analysis is based on bounding the error between the cont. time solution and the optimum and the error between the cont. time solution and its discrete counterpart. However, it is unclear how the discretization error varies as the step size and number of iterations vary.


The theorem also does not provide any basis to compare the three discretizations, since the same bound is provided, and the same set of assumptions is used for all discretization considered. An ideal analysis would likely distinguish different discretizations based on different assumptions (most likely smoothness of the loss function a la “Direct Runge-Kutta Discretization Achieves Acceleration”), and provide some meaningful non-asymptotic analysis of the magnitude of errors based on characteristics of the discretization.


The experiments section seems well-planned in general. Though the wall-time experiments could be inflating the gains vs Adam, since the first-order and second-order ema moments can be accumulated in parallel on a GPU (and this doesn’t require much effort to implement with standard frameworks, since most dispatches are async.).


Minor typos:
bottom of pg 5: “The analysis summarized in Theorem 2 is based on tools form hybrid control…” form -> from
middle of pg 6: “to achieve finite-time convergence in conitnuous-time” conitnuous-time -> continuous time

---

> ### Author Response · Authors · 2020-11-12
> **Thank you for your comments; please see our explanations below**
>
> {Q1- The submission analyzes convergence behavior of ... analyses which rely on differential inclusion.}:
>
> R1- Thank you for your time and for your feedback. We want to reiterate that this paper analyzes the convergence of three discretizations of the finite-time continuous q-RGF and q-SGF flows as proposed in Romero et al. 2020; please refer to that paper for details about the differences with the work of Wibisono et al. This being said, we agree that we could do a better job in explaining our motivation in targeting these flows, and the need of the notions of differential inclusions in the rigorous analysis of these, possibly discontinuous, flows.  The Introduction as well as the core of the paper have been amended to include some more explanations on this.
>
> {Q2- The second issue I’d like to raise is perhaps more important ...,and provide some meaningful non-asymptotic analysis of the magnitude of errors based on characteristics of the discretization.}:
>
> R2- Indeed, this point is important. However, we argue that the analysis results presented here, while perhaps preliminary, constitute an important and non-trivial analysis first step, in translating the continuous flows to their discrete counterparts. Indeed, this first result of `shadowing' the exact solutions of the continuous flows using the studied discretizations, as opposite to diverging away from the continuous solutions, is not a trivial task as this reviewer might agree. For instance, such an idea behind finding 'weak' bounds can be seen in other works on discretization of continuous flows, e.g., Barakat et al., arXiv:1810.02263, 2019 (https://arxiv.org/abs/1810.02263). Adding to this we show experimentally the performance of the proposed optimization algorithms on challenging DNN examples, as opposite to validations on trivial academic examples found in other works.
> On the point pertaining to Theorem 2, the notion of $(T,\epsilon)$-closeness is a notion of being close within an {\it arbitrarily small chosen $\epsilon$ } to the solutions of the continuous flows (see Definition 4, and Theorem 4 in Appendix). As such the result is useful in showing that the proposed discretizations will not diverge from the exact continuous solutions. However, we agree that the next step is to find an explicit relation between the step size $\eta$ and the chosen $\epsilon$; such relation will be of course different for the three discretizations. We have been working on this, however, as this reviewer knows very well, such results are not trivially obtained, and one cannot ask for every question to be solved within one conference paper.
>
> {Q3- The experiments section seems well-planned in general. ... accumulated in parallel on a GPU (and this doesn’t require much effort to implement with standard frameworks, since most dispatches are async.).}:
>
> R3- The numerical tests have been fairly conducted on the same numerical platform, to avoid confusions regarding what part helped in the acceleration of the convergence, i.e., hardware part or algorithmic part. Of course ADAM can be accelerated via particular hardware implementations, and so does the proposed methods. We believe that our comparison on a simple CPU platform allows us to correlate the potential convergence acceleration of the algorithm itself, without being confused with the extra degree of freedom allowed by a parallel implementation, which could be done in different ways, depending on what part of the algorithm is parallelized.
>
> {Q4- Minor typos: bottom of pg 5: “The analysis summarized in Theorem 2 is based on tools form hybrid control…” form -> from middle of pg 6: “to achieve finite-time convergence in conitnuous-time” conitnuous-time -> continuous time}:
>
> R4- Corrected, thank you.

---

> > ### Comment · AnonReviewer1 · 2020-11-14
> > **thanks for response; I maintain my original score**
> >
> > Thanks for the response.
> >
> > I will maintain my original score since the authors have not addressed my questions to the point where I think the major concerns are resolved.
> >
> > In particular,
> >
> > - from the response, the authors have not proposed any directions towards which the only theoretical contribution in the paper may be strengthened. In this regard, I do not think that the paper qualifies to be accepted as a theory paper. Moreover, the only claimed theoretical result does not fully support the main theme of the paper, which is the comparison of different discretization schemes of the flows considered.
> >
> > - "Of course ADAM can be accelerated via particular hardware implementations, and so does the proposed methods."
> >
> > My original point does not pertain to specializing to any particular hardaware implementation.
> >
> > With modern autodiff frameworks such as PyTorch, one may simple use the built-in optimizers directly along with a GPU so that non-interfering operations are executed (almost) in parallel. The way in which these frameworks are written makes running computations that do not block each other fairly easy -- one may just write the code for computation as they normally do to run programs on a CPU. See this note for example https://pytorch.org/docs/stable/notes/cuda.html
> >
> > Since such implementations are very widely used on GPUs, if not universally, it makes sense to compare the proposed method to Adam on GPUs if the message is to demonstrate a speed advantage in practice.
> >
> > For this reason, I do not think the paper qualifies for being accepted as an empirical paper.

---

> > > ### Author Response · Authors · 2020-11-15
> > > **Comment on the GPU vs. CPU implementation.**
> > >
> > > We are sorry to hear that this reviewer is still reluctant to give us a chance to meet and discuss this work with other colleagues at the conference.
> > > We might agree that the theoretical part is preliminary and hence the paper can be considered to be "not theoretical enough". However, we found that entirely dismissing the empirical value of the tests to be a rather harsh conclusion. Indeed, we are well aware of the existence of GPU implementations under Pytorch and other software platforms. We argue that GPUs are commonly used to speedup DNN applications in computer vision, speech, etc. This is, however, not the goal of this paper, where DNNs are merely used as additional examples, to show the performance of the algorithms {\it relatively to each other}, independently of the implementation platform. In other words, we believe that if we test the algorithms on the same platform, be it CPU, GPU, FPGA, etc. then the relative performance observed on one platform should hold true on other platforms, i.e, if we observed an acceleration of the proposed methods vs. others on CPU, and if all the methods are further equally accelerated via a GPU parallelization, then we expect the relative acceleration results between methods to hold true, even if the absolute computation time for every method will be faster on a GPU.
> > >
> > > Finally, we want to thank this reviewer for his comments, which we will certainly keep in mind for our future followup investigations on this subject.

---

### Official Review · AnonReviewer3 · 2020-10-28
**This paper provides several discretization strategies for three optimization flows. The main contribution comes from the convergence guarantee for the discrete system.**

**Rating:** 6
**Confidence:** 3

**Review:**

This paper provides several discretization strategies for three optimization flows. The main contribution comes from the convergence guarantee for the discrete system.
1. The convergence guarantee for continuous flow comes from previous work. Moreover, I understand that the author may wish to present the results as general as possible. However, since all the F is continuous and singleton in the main part (if I do not miss some parts), there is no need to introduce the differential inclusion based discontinuous dynamic system, which causes the appendix to be hard to read. All the auxiliary lemmas and theorems have a simple and intuitive version.
2. Eq. (32) is wrong. The order should be 1/(1-\alpha). Therefore, so do Theorem 2 also has a similar typo. At the end of page 15, "Therefore, Bala Bala...", it should be X \in D\\{x_0} not D\\{0}.
3. In the context of deep neural networks, transferring the convergence guarantee from the continuous optimization flow to the discrete system is quite important. However, this paper does not utilize any detailed information on DNNs; hence it is more like a general optimization paper. Considering there is rich work investigating the discretizations, I can't judge the contribution's significance of this paper.
4. In the experiment, the format and the resolution of the figures are not consistent.

---

> ### Author Response · Authors · 2020-11-12
> **Thank you for your comments; please see our explanations below**
>
> {Q1- The convergence guarantee for continuous flow comes from previous work. Moreover, I understand that the author may wish to present the results as general as possible. However, since all the F is continuous and singleton in the main part (if I do not miss some parts), there is no need to introduce the differential inclusion based discontinuous dynamic system, which causes the appendix to be hard to read. All the auxiliary lemmas and theorems have a simple and intuitive version.}:
>
> R1- Thanks for reading our paper and for your review work. We agree indeed, that the main discontinuity in the RGF/SGF flows happens at the equilibrium $x^{*}$, which makes the use of differential inclusions looks rather heavy-handed. However, in our previous work on the subject, we have been asked to study and show that all the intuitive concepts are still valid with the discontinuity at the equilibrium. As this reviewer pointed out, for completeness of the results, we reported the detailed proofs in Appendix, however, following your sensible suggestion, we added a sentence informing the reader that the notions/definitions about differential inclusions could be skipped  by the reader without loss of intuition behind the main results in the Appendix.
>
>
> {Q2-  Eq. (32) is wrong. The order should be $1/(1-\alpha)$. Therefore, so do Theorem 2 also has a similar typo. At the end of page 15, "Therefore, Bala Bala...", it should be $X \in D\{x_0\}$ not $D\{0\}$.}:
>
> R2- Yes, indeed. Typos corrected, thank you.
>
> {Q3-  In the context of deep neural networks, transferring the convergence guarantee from the continuous optimization flow to the discrete system is quite important. However, this paper does not utilize any detailed information on DNNs; hence it is more like a general optimization paper. Considering there is rich work investigating the discretizations, I can't judge the contribution's significance of this paper.}:
>
> R3- Yes, the paper is about proposing optimization algorithms via the discretization of the continuous optimization flows RGF/SGF. The DNN examples are used to show the performance of the optimization algorithms on challenging real-life examples. We have also removed from the Abstract the sentence `we investigate in the context of deep neural networks', since it might confuse the reader, giving the impression that we are tailoring the proposed optimization methods to DNNs, whereas we are using DNNs, among other examples, to validate these optimization methods.
>
> {Q4-  In the experiment, the format and the resolution of the figures are not consistent.}
>
> R4- Due to space limitation we had to make some of the figures smaller than what we wanted them to be. However, with the additional page allowed in the revised version, we have made some figures a little larger, improving the overall resolution.

---

### Official Review · AnonReviewer4 · 2020-10-29
**Review of continuous time methods**

**Rating:** 4
**Confidence:** 3

**Review:**



### Summary
This paper studied two continuous time methods called  rescaled-gradient flow and the signed gradient flow and proposed three efficient discretization, namely the forward-Euler, Runge-Kutta and Nesterov discretization. The paper demonstrated the finite-time convergence of the proposed methods under gradient dominance assumption, and use experiments to show that the proposed methods outperform standard stochastic gradient algorithms in many deep learning tasks.

### Comments

The paper is clearly structured; the related work in the dynamical systems and control are well presented; experiments are detailed and rigorous, it is good to see detailed report on hyperparameters in appendix.


While the paper claims that the study is in the context of deep learning, it is unclear to me whether the theoretical analysis is well suited for deep learning models. In particular, the algorithm and main convergence result (Theorem 2) require full gradient. It would be more interesting to show  convergence result for stochastic setting where only mini batch of samples are provided each time.

In abstract, the paper claims the flows include  non-Lipscthiz or discontinuous system, can the paper give a few examples of such cases? If I understand correctly,  the gradient in example 1 is continuous due to Lipschitz continuity, and the proximal mapping in example 2 is continuous due to nonexpansiveness.


The main result Theorem 2 seems to be problematic. It claims that after finite number of iteration, the iterates $x_k$ will eventually fall in a small $\epsilon$ neighborhood for some $\epsilon$ error. However, it does not specify how small $\epsilon$ is. By using sufficiently large $\epsilon$, the weak bound is always valid. Can we prove the weak convergence bound (eq. 21) for arbitrarily small  $\epsilon$?

The main convergence property is established on the gradient dominance condition (13), which seems to be relatively strong because it requires isolated saddle points. Can the authors elaborate the intuition of (13) for deep learning models?

In the experiment (Figure 8, appendix) both training loss and testing loss are plotted. However,  it appears that testing loss is much smaller than the training loss. Is it normal or does it have any underfitting issue?

Since the main focus (theory part) is on deterministic setting, to confirm the theoretical finding and  to show the advantage of continuous time methods,  it would be more interesting to compare with Nesterov method or other gradient method for deterministic optimization.  I feel that this work would be more interesting to the field of dynamic control or optimization rather than deep learning.

Typo:
In 4.2.2.
"Nesterov's Nesterov accelerated gradient descent" should be "Nesterov's  accelerated gradient descent"

---

> ### Author Response · Authors · 2020-11-12
> **Thank you for your comments; please see our explanations below.**
>
> {Q1- This paper studied two continuous time methods... in many deep learning tasks.}:
>
> R1- First, thank you for taking the time to read our paper and for your review. We want to underline that the paper is not about the continuous version of the optimization flows, which have been introduced in the paper Romero et al. 20. This paper is about proposing three discretization methods for the q-RGF and q-SGF flows, analyzing their stability, and showing their performances on academic examples, as well as DNN examples.
>
> {Q2- While the paper claims that the study is in the context of deep learning, it is unclear to me ... only mini batch of samples are provided each time.} :
>
> R2-The optimization methods and their analysis in the form of Theorem 2, have been proposed in the deterministic setting. Similarly, all our academic examples ( we reported two examples in the paper; one in the main paper and one in the Appendix) have been done in the deterministic setting as well. After that, we decided to push the tests further by trying the proposed optimization methods on real-life challenges in the context of DNNs. First, we tested all the algorithms in the deterministic setting and the qualitative results were the same, as the one finally reported in the paper. However, some colleagues, who are specialists in DNNs, suggested that it would be perhaps more convincing to compare the proposed methods against the fastest and best optimization methods on DNNs, which happen to be in the stochastic setting, e.g., SGD. For this reason, we then re-did all the tests in the stochastic setting, for all methods, and found similar conclusions, qualitatively. Based on that we reported the latter results in the paper. We added some explanations about this testing process in the paper. We also agree that going back and analyzing the convergence of the proposed methods in the stochastic setting is important, more so now that we observed their performances, and this will be one of our future research focus.
>
> {Q3- In abstract, the paper claims the flows include non-Lipscthiz or discontinuous system... in example 2 is continuous due to nonexpansiveness}:
>
> R3- As indicated in the Abstract, the non-Lypschitz and discontinuous are meant for the q-RGF and q-SGF only. The examples 1-3 are simply introduced to explain the formulation of optimization methods in the context of the state-space form (6).
>
> {Q4- The main result Theorem 2 seems to be problematic. ... convergence bound (eq. 21) for arbitrarily small $\epsilon$} :
>
> R4- The results of $\epsilon$-closeness are meant in the sense for arbitrarily small $\epsilon$ (see Definition 4, and Theorem 4 in Appendix). The result of Theorem 2, is an existence result, in the sense that we show, for an arbitrarily small $\epsilon$, the existence of sufficiently small $\eta$ such that $\epsilon$-closeness holds between the continuous-time solutions and the discrete-time solutions. However, even though this first result is interesting and challenging on its own, we agree that this is still 'weak' bound, and for it to be a 'strong' bound, we need to further find an explicit characterization of $\eta$ as function of $\epsilon$. This is an ongoing, far from trivial, effort.
>
> {Q5- The main convergence property is established on ... of (13) for deep learning models?}:
>
> R5- There is evidence that gradient dominance does hold locally in many deep learning contexts (Zhou and Liang, 2017, https://arxiv.org/abs/1710.06910). Indeed, since convexity readily leads to gradient dominance of order $p = \infty$, it suffices that a slightly stronger form of it holds (but weaker than strong convexity), in order to have $p<\infty$, and thus for us to be able to choose $q>p$; see  (Remark 2, Appendix E) in the updated version of the paper.
>
> {Q6- In the experiment (Figure 8, appendix) both training loss and testing loss are plotted. ... does it have any underfitting issue?}:
>
> R6- Those are the results obtained for the MNIST, with CNN models, since our goal is the performance on the test set to be good (about $99\%$), we did not find that a slightly smaller test loss (around $0.12$) compared to the training loss (around $0.2$) constituted a major problem in this case.
>
> {Q7- Since the main focus (theory part) is on deterministic setting,... dynamic control or optimization rather than deep learning.}:
>
> R7- Please see our response above regarding this point.
>
> {Q8- Typo: In 4.2.2. "Nesterov's Nesterov accelerated gradient descent" should be "Nesterov's accelerated gradient descent"}:
>
> R8- Corrected, thank you.

---

### Decision · Program_Chairs · 2021-01-07
**Final Decision**

**Decision:**

Reject

**Comment:**

The paper proposes three discretization schemes for two first-order optimization flows, and proves the "convergence" to the minimizers of the problem that the optimization flows approach. The methods are tested on the DNN training problem and show comparable performance.

Pros:
1. The problem being studied, the discretization of optimization flows, is of interest to the community.
2. "Convergence" guarantee is provided.

Cons:
1. The theoretical analysis is somewhat preliminary, as the authors have admitted. There is a prescribed \epsilon in the approximation error (23) that prevents the right hand side of (23) from approaching zero. The parameter \eta, depending on the chosen accuracy \epsilon, should be provided so that a user can implement the discretization schemes if s/he is interested. Moreover, by specifying \eta, it may be possible to compare the numbers of iterations to approach an \epsilon-solution between the proposed discretization schemes and other optimization methods for solving the original optimization problem. By doing this, the motivation issue from Reviewer #1 (and the AC) can be resolved. Purely discretizing an optimization flow is of less interest to the machine learning community.
2. Although the comparison on academic problem is obviously advantageous, the comparison on DNN training is only comparable or marginally better.

The author responses resolved part of the challenges from the reviewers, but the key issues remained (as communicated in confidential comments). Since the final average score is below threshold, the AC decided to reject the paper.